# Modeling glioblastoma heterogeneity as a dynamic network of cell states

Ida Larsson[1,†] (ID), Erika Dalmo[1,†] (ID), Ramy Elgendy[1] (ID), Mia Niklasson[1] (ID), Milena Doroszko[1] (ID), Anna Segerman[1,2] (ID), Rebecka Jörnsten[3], Bengt Westermark[1,‡] (ID) & Sven Nelander[1,*,‡] (ID)

## Abstract

**Tumor cell heterogeneity is a crucial characteristic of malignant brain tumors and underpins phenomena such as therapy resistance and tumor recurrence. Advances in single-cell analysis have enabled the delineation of distinct cellular states of brain tumor cells, but the time-dependent changes in such states remain poorly understood. Here, we construct quantitative models of the time-dependent transcriptional variation of patient-derived glioblastoma (GBM) cells. We build the models by sampling and profiling barcoded GBM cells and their progeny over the course of 3 weeks and by fitting a mathematical model to estimate changes in GBM cell states and their growth rates. Our model suggests a hierarchical yet plastic organization of GBM, where the rates and patterns of cell state switching are partly patient-specific. Therapeutic interventions produce complex dynamic effects, including inhibition of specific states and altered differentiation. Our method provides a general strategy to uncover time-dependent changes in cancer cells and offers a way to evaluate and predict how therapy affects cell state composition.**

**Keywords** cell state; cellular barcoding; patient-derived brain tumor cells; single-cell lineage tracing; time-dependent computational models
**Subject Categories** Cancer; Methods & Resources; Neuroscience
**Mol Syst Biol. (2021) 17: e10105**

## Introduction

During the last two decades, there has been a multifaceted debate on the time-dependent variation of differentiation states in tumor cells. According to the cancer stem cell model, both solid tumors and leukemias tend to follow developmental hierarchies, with stem-like cells at the apex (Reya *et al*, 2001; Pardal *et al*, 2003; Singh *et al*, 2004. A complementary set of ideas, often referred to as tumor cell plasticity, emphasize a less structured variation, whereby tumor cells can switch multi-directionally between stem-like and differentiated states, or between migratory and proliferative states, tending toward stochastic equilibrium (Gupta *et al*, 2011; Gerlee & Nelander, 2012; Dirkse *et al*, 2019). Crucially, the two perspectives have different therapeutic implications; whereas the cancer stem cell model logically implies that targeting the stem-like cells might eradicate the tumor (Dingli & Michor, 2006), the latter (plasticity) model suggests that tumor growth and therapeutic responses will depend in a complex and quantitative manner on the specific switching patterns and rates (Gupta *et al*, 2011; Gerlee & Nelander, 2012). This study aims to establish a method that can resolve and quantify the time-dependent heterogeneity of tumor cells, both during normal growth and during treatment.

Glioblastoma (GBM) is a malignant brain tumor characterized by short survival and a lack of effective therapeutic options (Lesueur *et al*, 2019). Patient tumors can be divided into three gene expression subtypes termed proneural, mesenchymal, and classical (Wang *et al*, 2018), but the heterogeneity is not restricted to variation between patients. Already in the 1930s, it was noted that GBMs exhibit a high degree of cell-to-cell heterogeneity, as evident from distinct histological structures within the tumor (Scherer, 1938; Puchalski *et al*, 2018). Recent explorations of GBM by single-cell RNA profiling have demonstrated that GBM cells exist in a variety of transcriptional states, indicating differences in neuronal and glial differentiation, as well as differential activity of gene programs involved in mesenchymal transformation and cell cycling (Patel *et al*, 2014; Neftel *et al*, 2019). It is expected that this diversity is a key factor underlying tumor recurrence and response to therapy (Bedard *et al*, 2013), but despite our increasing ability to detect GBM differentiation states, their time-dependent variation remains poorly understood. Whereas xenograft studies have supported a hierarchical organization, driven by a subpopulation of CD133+ tumor-initiating cells (Singh *et al*, 2004; Lan *et al*, 2017), several lines of evidence, including time-lapse microscopy, RNA velocity measurements, and *in vivo* lineage tracing, support a less hierarchical (plastic) organization (Farin *et al*, 2006; Gerlee & Nelander, 2012; Dirkse *et al*, 2019; Neftel *et al*, 2019; Wang *et al*, 2019; Couturier *et al*, 2020). Thus, further clarifying the dynamics of

---

1   Department of Immunology, Genetics and Pathology, Uppsala University, Uppsala, Sweden
2   Department of Medical Sciences, Cancer Pharmacology and Computational Medicine, Uppsala University Hospital, Uppsala, Sweden
3   Mathematical Sciences, Chalmers University of Technology, Gothenburg, Sweden
    *Corresponding author. Tel: +46 76 1380123; E-mail: sven.nelander@igp.uu.se
    †These authors contributed equally to this work as first authors
    ‡These authors contributed equally to this work as senior authors

transcriptional states in GBM will have important consequences for therapy development and our understanding of the disease.

Herein, we describe a quantitative strategy to uncover both hierarchical variation and plastic variation of GBM differentiation states, based on the integration of molecular barcodes, single-cell profiling, and mathematical modeling. Applied to cells from a selected set of patient-derived GBM cell lines, our proposed state transition and growth (STAG) model achieves *de novo* identification of transcriptional states and provides estimates of state-specific growth rates and the frequencies of transitions between the states. The fitted STAG models are hierarchically structured networks (including a top state) with multi-directional switching between intermediate states that recapitulate specific neural cell types. To demonstrate how our strategy can account for the effects of therapies, we extend our model to measure how treatment with a clinically used drug (temozolomide) and a growth-inhibiting cytokine (BMP4) change the cell state transitions. We further propose a mathematical criterion to predict how drugs affect population growth and state composition, based on the eigendecomposition of the matrix of cell state transition rates.

Altogether, our results introduce a new method to model mathematically the cell state changes in solid tumor cells under normal growth and therapeutic intervention. We expect that our model will have interesting applications in the assessment of novel therapies and in the formulation of strategies to enhance the effects of standard therapies.

# Results

The goal of the developed method is to measure how the transcriptional state of individual GBM cells changes over time and to estimate how such changes are affected by treatment. Previous reports (Gupta *et al*, 2011; Dirkse *et al*, 2019; Neftel *et al*, 2019) have demonstrated that single cells, or purified populations of tumor cells (consisting of a single state), can give rise to a mixture of states, implying that transitions between cell states have occurred. Here, we present a more general method that does not require the purification of cells in any particular state. Our strategy comprises three main steps (Fig 1A). First, we introduce a set of unique heritable barcodes into a culture of patient-derived cells. Next, we propagate the cells in culture, sampling a fraction of cells at regular time intervals and use single-cell RNA sequencing to determine both the transcriptome and the barcode of each cell. Our algorithm, STAG, is subsequently used to build a model of the data, which identifies the cell states, their growth rates, and the structure and rates of the

transitions. With particular experimental and computational extensions, described below, STAG can also measure how specific drugs alter the cell state transitions and predict interventions most likely to reduce net tumor growth.

## Lineage tracing and profiling of barcoded glioblastoma cells

As a first test case for our approach, we chose the well-characterized GBM cell culture U3065MG, derived from a 77-year-old male patient. U3065MG cells classify as mesenchymal subtype, are *TP53* and *IDH1* wild type, form infiltrative macroscopic tumors *in vivo*, and harbor a subset of clonogenic cells that can give rise to both primary and secondary cultures (Xie *et al*, 2015; Segerman *et al*, 2016). We transduced passage 6 U3065MG cells with a lentiviral mRNA barcode library (Adamson *et al*, 2016) at a low (0.1) multiplicity of infection to ensure that a majority of transduced cells would carry a single unique barcode. An initial population of 2,500 barcoded cells was subsequently grown for 21 days *in vitro*. Cells were passaged each week; on each passage, we harvested 80% of the culture for single-cell RNA sequencing and kept the remaining cells (20%) in culture. This way, we obtained a series of observations for cells on days 7, 14, and 21. The fraction of sampled cells (80%) was chosen to guarantee a sufficiently big random sample of cells with each barcode, and the time scale of the experiment (21 days) aimed to capture gradual changes in cell differentiation state (see Appendix for details). As a reference point, to validate the barcoding, we also included a separate sample of cells immediately after barcoding (0 days). No apparent batch effects or skewness of the transcriptome due to the barcoding could be found (Appendix Fig S1).

We first analyzed the number of cells carrying a particular barcode, i.e., the clone size. As expected, the clone size was precisely 1 immediately after barcoding. As the experiment progressed, the variation of observed clone sizes increased (Fig 1B), and the number of unique remaining clones decreased, suggesting the extinction of some clones (Fig 1C). To account for these trends, we applied a stochastic model of clonal growth with fixed cell proliferation and death rates (formally, a Galton–Watson process (Watson & Galton, 1875)) for each clone. A simulation of such a process fitted to our data recapitulated with good accuracy the experimental observations of dispersing clone sizes and the number of observed barcodes per time point (Fig 1D and E).

To explore biological factors that might influence clone size, we analyzed the differential gene expression between cells belonging to large vs small clones using gene set enrichment analysis (GSEA). Small clones were characterized by, e.g., up-regulation of the

**Figure 1. Resolving the plasticity of GBM cells.**

A Overview. In the STAG procedure, barcoded tumor cells are sampled at multiple time points and profiled by single-cell RNA sequencing, followed by mathematical modeling to identify (i) the states and their growth rates, (ii) the patterns and rates of the state transitions, (iii) how drugs affect cell states, and (iv) analysis of cell population stability and long-term projections of cell state compositions.

B Clone sizes of barcoded U3065MG cells at 0–21 days. *X*-axis, barcodes; *Y*-axis, number of cells containing each barcode.

C Venn diagram of number of barcodes detected at 7, 14, and 21 days.

D, E Simulation of the experiment in (B,C) using a Galton–Watson process with a fixed growth and death rate.

F Enriched gene sets in small and large clones, respectively, using Gene Set Enrichment Analysis (GSEA). Normalized enrichment scores (NES) and *q*-values are indicated in the figure.

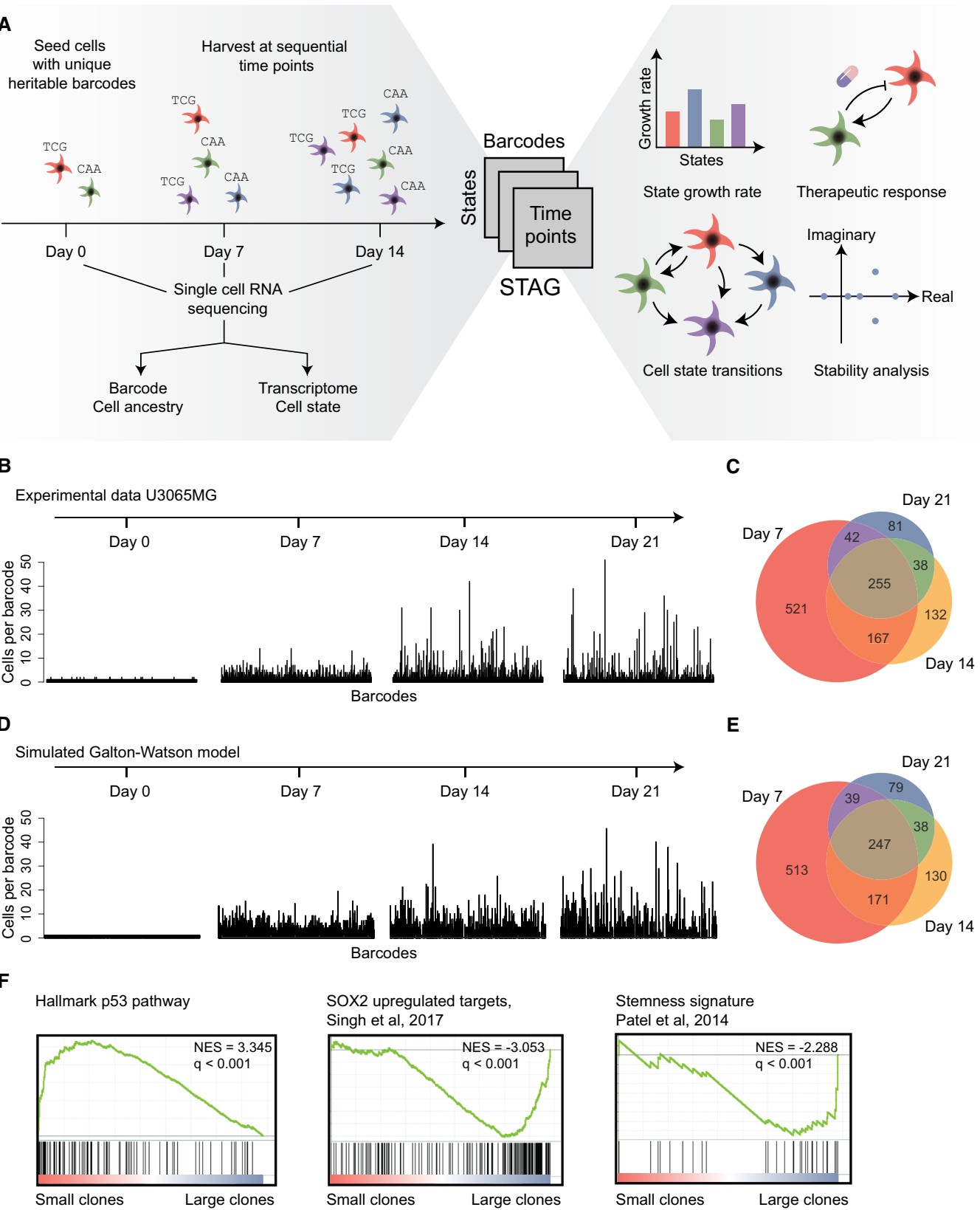

**Figure 1.**

## Box 1.  The State Transition and Growth (STAG) model

**Key model concepts and equations.** The state transition and growth model is based on 4 rules:

1. tumor cells exist in either of *k* states,
2. each of the *k* states has a specific proliferaton and death rate,
3. tumor cells stochastically transit between the states at some fixed average rate and
4. an externally applied drug can alter the transition rates, the growth rates or both.

The transition rates in (3) are constrained to be non-negative to estimate transitions in both directions. The system is approximated by continuous first order rate equations, whereby all the transition and growth rates are summarized by a $k * k$ matrix, *A*, which gives the explicit solution

$$X(t_s) = e^{At}X(t_{s-1}) \qquad (1)$$

where $X(t_s)$ and $X(t_{s-1})$ are the state distribution over barcodes at time $t_s$ and $t_{s-1}$, respectively, and *e* denotes the matrix exponential. To illustrate the transition matrix *A* and the parameters that can be derived from it, we use the specific case of *k = 2*:

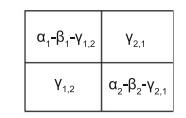

Using a first degree approximation of the matrix exponential (Materials and Methods), the sum-of-squares error over all barcodes is given by

$$E(X, A) = \sum_{s=2,3} ||X(t_s) - \tilde{A}(t_s - t_{s-1})X(t_{s-1})\tilde{\eta}_s||_{Fro} \qquad (2)$$

where $\tilde{\eta}_s$ is an adjustment factor for harvesting of cells. *E(X,A)* is minimized using convex optimization to find *A*, given the experimental data.

To capture the treatment-specific component of the transition network, the transition matrix was redefined as consisting of one joint and one treatment-specific component, expressed as

$$A_{treatment} = A_{untreated} + \Delta A_{treatment} \qquad (3)$$

$A_{untreated}$ represents the baseline network, i.e. that of untreated cells, and $\Delta A_{treatment}$ the changes in transition and growth rates due to treatment:

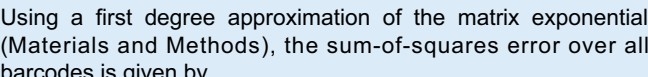

The *A*-matrix can be further used to predict the minimal intervention needed to stabilize the tumor cell population, i.e. suppress growth. We note that if we let *X(t)* denote the total number of tumor cells in all states, i.e. $X(t) = X_1 + X_2 +,...,+ X_k$, the matrix *A* fully denotes how *X(t)* will change over time. Theory for linear dynamical systems (Varfolomeev & Lukovenkov, 2018) implies that *X* will have a steady state if and only if

$$Re(\lambda_i) \leq 0, \ i = 1, 2, ..., k \qquad (4)$$

where $\lambda_i$ are the eigenvalues of *A*. We can compute the eigenvalues of *A* to determine if the net growth is stable. Furthermore, we can predict a minimal intervention $\Delta A$ that stabilizes the cell population by formulating a convex optimixation problem and solve for the minimal change needed to the *A*-matrix to obtain a stable system, which satisfies the convex relaxation of stability, proposed in (Zavlanos *et al*, 2011):

$$a_{ii} \leq -\sum_{j \neq i} |a_{ij}|, i = 1, ..., n \qquad (5)$$

Hallmark p53 pathway, consistent with p53 being a tumor suppressor, whereas the larger clones upregulated SOX2 and stemness signatures (Fig 1F), consistent with these clones having a higher proliferation rate. We conclude that our barcoding strategy can trace GBM clonal growth with accuracy and that the transcriptional profile and clonal growth rate were correlated, warranting further investigation.

### The state transition and growth model

Next, we developed a computational model to integrate variation in cell state and cell growth. Mathematically, a natural way to represent state transitions in cells is by Markov chain modeling (Gupta *et al*, 2011; Dirkse *et al*, 2019). Markov chains can represent both hierarchical switching and multi-directional switching, but do not account for cell proliferation and death. This is problematic since, for instance, the rapid proliferation of one state may be misinterpreted as transitions toward that state. We, therefore, considered a model based on Gerlee and Nelander (2012), in which growing GBM cells alternate between 2 states. This 2-state model, however, is not consistent with recent data suggesting multiple states in GBM cells (Neftel *et al*, 2019). Here, we propose a more general class of models (Box 1) to integrate cellular state transitions and growth, based on four assumptions:

- Tumor cells exist in either of *k* transcriptional states.
- Each of the *k* states has specific proliferation and death rates.
- Tumor cells randomly transit between states at specific rates.
- An externally applied drug can alter the transition parameters, the growth parameters, or both.

Together, these rules describe a stochastic model which can be simulated, e.g., using Gillespie's method (Gillespie, 1976). Our model contains previous models as special cases; for instance, when

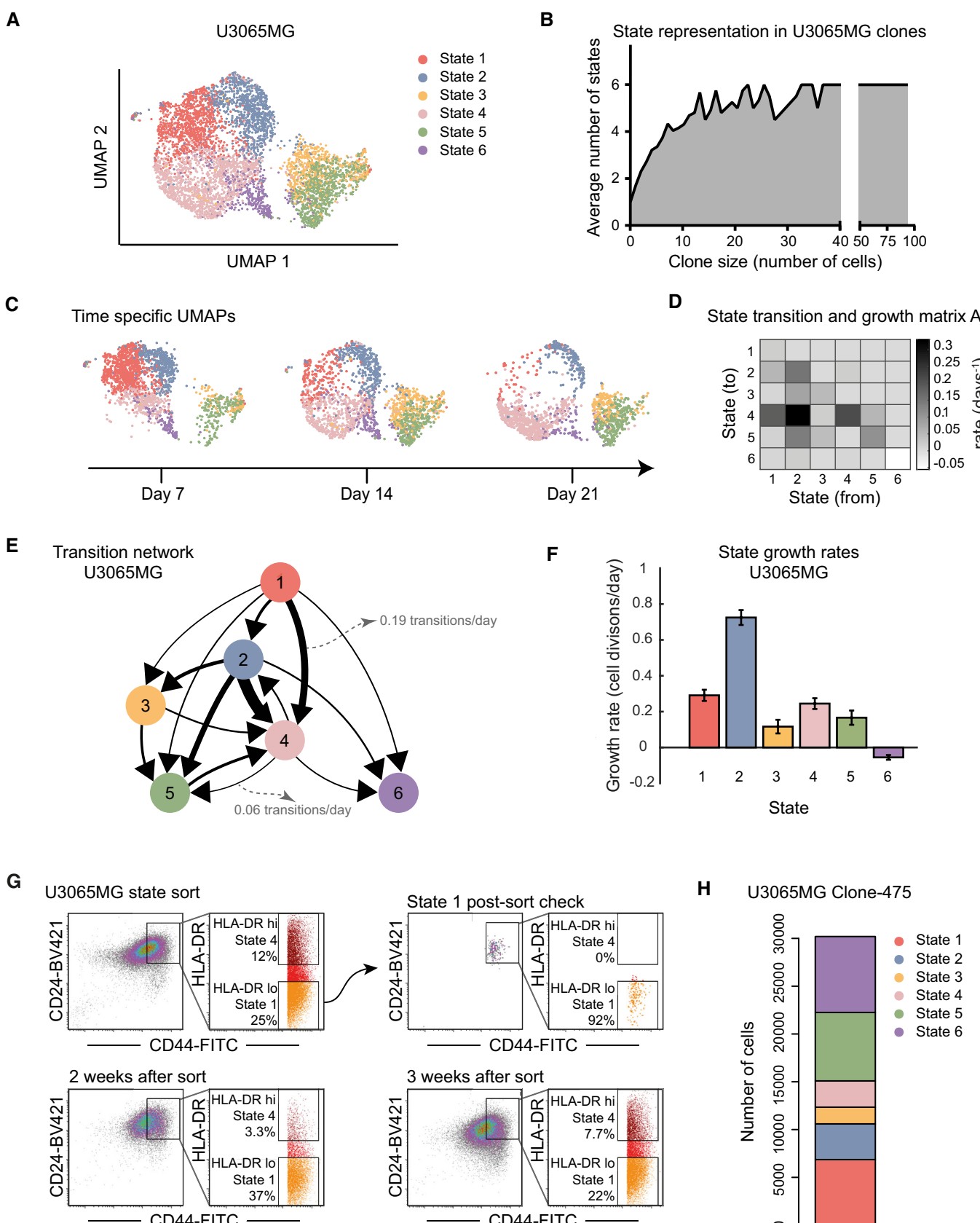

**Figure 2.**

**Figure 2.  State dynamics of cells from the mesenchymal U3065MG GBM cell line.**

A   UMAP embedding of single U3065MG cells colored according to transcriptional cell state assignment.
B   Average number of states represented in each clone with increasing clone size. *X*-axis, clone size; *Y*-axis, average number of states.
C   UMAP embedding of cell state distribution at each experimental time point. State colors as in (A).
D   Heatmap of the U3065MG state transition and growth matrix A derived from the STAG model.
E   Estimated network of state transitions in the U3065MG cell line. The thickness of the arrows correlates with the rate of transition (number of transitions per day).
F   STAG estimates of growth rates (cell divisions per day) for U3065MG states 1–6. Error bars indicate the 90% confidence interval, based on 1,000 bootstrap replicates.
G   Enrichment of the U3065MG state 1 cell population (CD24$^{high}$/CD44$^{high}$/HLA-DR$^{low}$) using FACS cell sorting. Cells were monitored by flow cytometry over 3 weeks, and a gradual phenotypic drift, e.g., toward state 4 (CD24$^{high}$/CD44$^{high}$/HLA-DR$^{high}$), could be observed. See also Fig EV1.
H   scRNA sequencing of a U3065MG-derived clonal culture (clone 475) detected all six states of the U3065MG mother cell line. U3065MG-C475 cells were scored against U3065MG state 1–6 signatures using ssGSEA.

$k = 1$ (a single state), it corresponds to a standard stochastic model of clonal growth (Galton–Watson process, as above); when states represent migratory and proliferative GBM cells ($k = 2$), it is equivalent to the Gerlee and Nelander (2012) two-state model; and when the model has multiple states and the proliferation and death rates are zero, it is equivalent to the Markov chain model (Gupta *et al*, 2011; Dirkse *et al*, 2019).

Note that our model makes no prior assumptions regarding the number and type of states, or how the states are connected. Based on the data, it can be used to represent either a hierarchical organization, as well as one characterized by multi-directional transitions.

### Fitting state transition and growth models to experimental data

Given the above STAG model, we developed an efficient method for fitting its parameters from barcoded single-cell RNA profiling data, as follows. We first note that all the transition and growth rates are summarized by a $k$ times $k$ matrix, $A = a_{ij}$, which can be thought of as a network map of the cell state transitions (Box 1). Specifically, the non-diagonal elements $a_{ij}$, $i{\neq}j$ of $A$ represent the transition rate from state $j$ to state $i$. The column sums of $A$, in turn, represent the net growth of each state (Box 1). Given observational data, our algorithmic goal is to obtain estimates of all the elements of $A$. We do this by a convex optimization algorithm, which minimizes a global error function over the full data set, comprising the state distributions of each barcode at days 7, 14, and 21 (Materials and Methods, Equations 6 and 7). Due to the convex formulation, a complete and unique solution can be rapidly obtained (fractions of a second). To benchmark our algorithm, we performed a large number of simulations, in which STAG models of different sizes ($k$) and connectivity were simulated under different experimental designs. The simulations supported that the particular experimental setup, a 3-week experimental design with sampling every 7 days and a 6-state system (discussed below), could identify the elements of $A$ (i.e., the transitions and growth rates) with low error (Appendix Fig S2).

### State transitions and growth dynamics of U3065MG cells

To construct a STAG model of U3065MG cells, we first divided all profiled cells into discrete cell states, based on their RNA profiles. We assigned states by consensus *k*-means clustering (Fig 2A, Dataset EV1). The number of states ($k = 6$) was selected by boot-strapping methods (Appendix Fig S3). Relating states to clone sizes, we found all six states present in clones with at least 37 detected

cells (Fig 2B). Moreover, cells from all six states could be found at each time point (Fig 2C).

The best-fitting STAG model derived from the data contained transitions ranging from 0.01 to 0.3 transitions per day (Fig 2D). The STAG network of detected transitions in U3065MG cells was hierarchically organized (Fig 2E) with substantial differences in growth rates of each state (Fig 2F). For instance, the top state (1) grew at 0.29 divisions/day and had only outgoing transitions. Immediately downstream, state 2 had the fastest growth rate (0.79 divisions/day), followed by state 3–5 with growth rates between 0.12 and 0.25 divisions/day. We found several cases of multi-directional switching between states 2–5. At the bottom of the hierarchy, state 6 had a negative growth rate and not outgoing transitions and thus acted as a sink state in the STAG network.

We performed two additional experiments to confirm that transitions occur in the U3065MG cells. In the first, we isolated state 1 cells by fluorescence-activated cell sorting (FACS), based on the surface markers CD24, CD44 (high in states 1 and 4), and HLA-DR (low in state 1, high in state 4; Fig EV1A). We observed a gradual phenotypic drift away from the purified state 1 cell population into, e.g., state 4 (CD24high/CD44high/HLA-DRhigh; Figs 2G and EV1B). Second, we analyzed a U3065MG-derived clonal culture, U3065MG-C475 (Segerman *et al*, 2016), by single-cell RNA sequencing and assigned each cell one of the six states using single-sample GSEA (ssGSEA). We found that the clonal U3065MG-C475 culture contained all six states found in the parental U3065MG culture (Fig 2H). These experiments further support that cell state transitions take place in the U3065MG cell culture.

### U3065MG cell states have distinct functional signatures

We carried out GSEA to obtain a functional profile for each state, based on its differentially expressed genes (Fig 3A), considering both the MSigDB Hallmark pathways and a collection of gene signatures relevant to GBM biology (Wang *et al*, 2018; Zhong *et al*, 2018; Neftel *et al*, 2019; Weng *et al*, 2019; Couturier *et al*, 2020; Garofano *et al*, 2021). The GSEA identified the top state 1 as the most mesenchymal state, matching signatures by, e.g., Neftel *et al* (2019) and Wang *et al* (2018). State 2, in turn, was enriched for proneural markers and neural progenitor profiles (Zhong *et al*, 2018; Garofano *et al*, 2021) and an up-regulation of cell cycle-related gene sets. Thus, state 2 resembles a rapidly proliferating progenitor with a proneural profile. States 1 and 2 were both enriched for markers of the glial progenitor-like cells defined by Couturier *et al* (2020), consistent with a position of these states at the top of the hierarchy.

**A**

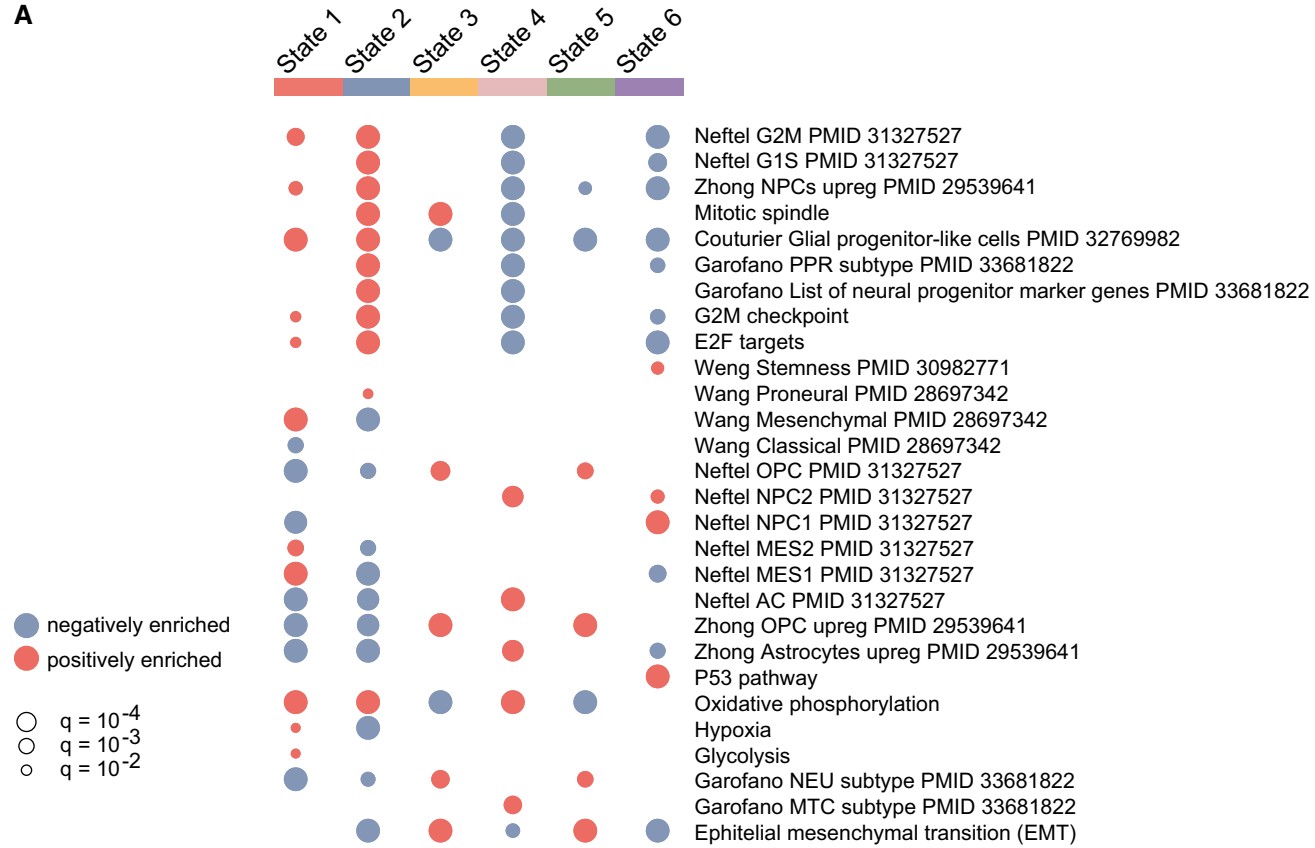

**B**

Survival analysis of GBM patients in TCGA cohort

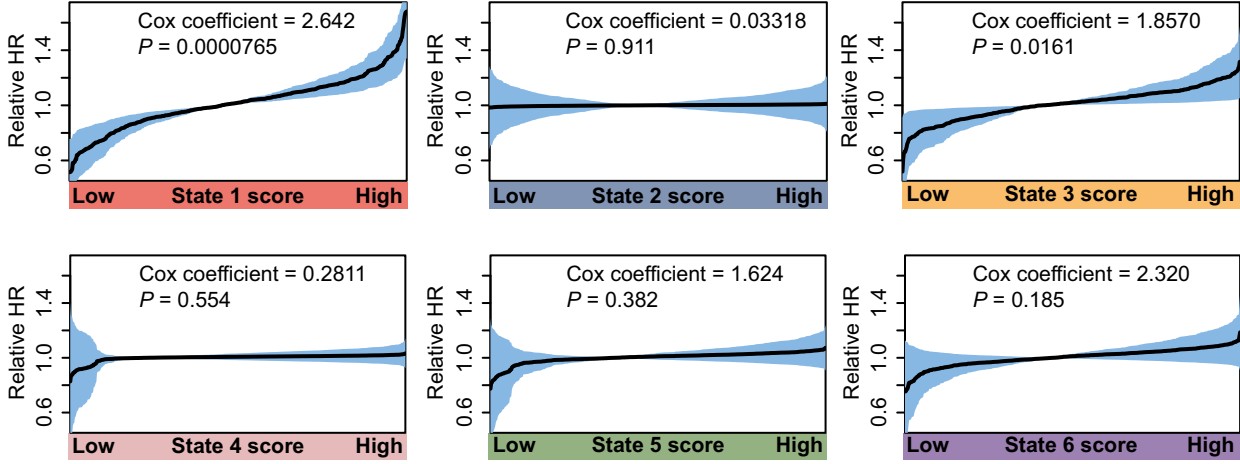

**Figure 3.  Characterization of states in the mesenchymal U3065MG GBM cell line.**

A  GSEA of the gene signatures for each of the six U3065MG states. Blue and red indicate negative and positive enrichments of the pathway. Size of the dot indicates significance (proportional to the GSEA false discovery rate [FDR] q-value). Gene sets without PMID reference were obtained from the MSigDB Hallmarks database of gene sets. Abbreviations: oligodendrocyte progenitor cell (OPC), neural progenitor cell (NPC), mesenchymal (MES), astrocyte (AC), neuronal (NEU), mitochondrial (MTC), proliferative/progenitor (PPR).

B  Survival analysis of TCGA GBM patients estimated by Cox's proportional hazards model, with enrichment score of states 1–6 as independent covariate. Shaded areas indicate 95% confidence intervals, calculated as ± 1.96 * standard error (SE)). HR, hazard ratio.

Source data are available online for this figure.

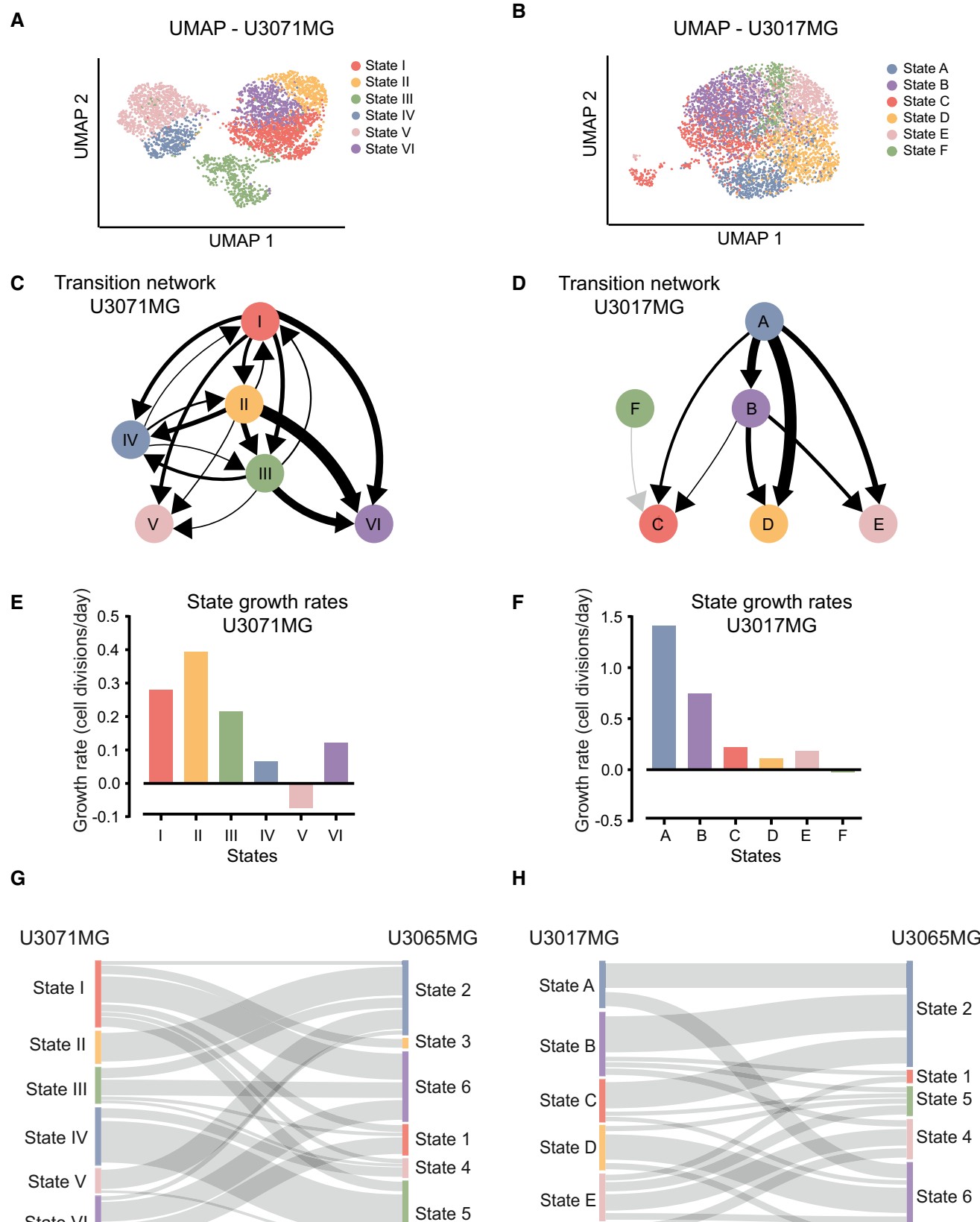

**Figure 4.**

◀

**Figure 4.   State dynamics of cells from the mesenchymal U3071MG and the classical U3017MG GBM cell lines.**

A, B   UMAP embedding of single cells from the mesenchymal U3071MG (A) and classical U3017MG (B) GBM cell lines, colored according to transcriptional state assignments I–VI and A–F, respectively.

C, D   Estimated network of state transitions in U3071MG (C) and U3017MG (D). The thickness of the arrows correlates with the rate of transition (number of transitions per day, as in Fig 2E).

E-F   State growth rates (number of cell divisions per day) in U3071MG (E) and U3017MG (F).

G, H   Sankey plot of the relation between the states defined in U3065MG cells (1–6), and the states defined in U3071MG cells (I–VI) (G) and in U3017MG cells (A–F) (H). U3071MG and U3017MG cells were scored against U3065MG state 1–6 signatures using ssGSEA.

The states 3 and 5 matched oligodendrocyte precursor cell (OPC) signatures described by Neftel *et al* (2019) and Zhong *et al* (2018) as well as the Hallmark mesenchymal transition signature. Although states 3 and 5 were similar, enrichment for the mitotic spindle was found in state 3, which was absent in state 5. Also when cluster number was reduced, states 3 and 5 remained separate (Fig EV2), motivating the interpretation of states 3 and 5 as two separate cell states. State 4 matched astrocyte signatures described by Neftel *et al* (2019) and Zhong *et al* (2018). However, its enrichment for neural progenitor cells (Neftel NPC2) and the ability of state 4 cells to transit up in the hierarchy toward state 2 indicate that this state does not represent terminally differentiated astrocyte-like cells. Last, state 6 was enriched for a Hallmark p53 response, and a majority of the top marker genes of state 6 have functions linked to the inhibition of cell proliferation, cell cycle arrest, and the apoptosis, such as *CDKN1A*, *GADD45A*, and *SOX4*, which helps explain the negative growth (Fig 2F) of state 6 (Dataset EV1).

We further related our state signatures to The Cancer Genome Atlas (TCGA) patient cohort through ssGSEA. Cox's proportional hazards models were constructed using the enrichment score for each state in each patient as the predictor variable. For U3065MG, this revealed a significant association between a high state 1 score and poorer survival (Cox coefficient = 2.642, $P < 0.001$; Fig 3B). A weaker association was found for a high state 3 score (Cox coefficient = 1.857, $P = 0.016$) while remaining states were not significantly associated with survival.

## Distinct hierarchical STAG networks in cells from different patients

To address the generality of our findings, we performed RNA profiling and STAG analysis using two additional patient-derived GBM cultures. One, U3071MG (male patient, 65 years old), was a mesenchymal cell line chosen to be similar to U3065MG. The other, U3017MG (female, 68 years), has classical subtype designation and was selected to be distinct from the other two (Appendix Fig S4). Like for the U3065MG cells, a subdivision into six states was preferred for both U3071MG and U3017MG, as judged by consensus clustering (Fig 4A and B; Appendix Figs S5 and S6). In the below presentation, we denote the U3071MG states by Latin numerals (I–VI) and the U3017MG states by letters (A–F).

The STAG network detected for U3071MG was hierarchically structured, in a way that resembled U3065MG, although with a higher degree of bi-directionality in its transitions (Fig 4C). For instance, a degree of influx into the top state (state I) could be detected, which was absent in the U3065MG cell line. In addition, two sink states (states V and IV) with no outgoing transitions were found. The states displayed substantial differences in growth rates, and, just as for the U3065MG STAG network, the fastest growing

state was found second to the top in the hierarchy (state II) and the only state with a negative growth rate was identified as one of the sink states (state V; Fig 4E).

The U3017MG STAG network had no bi-directional switching between states (Fig 4D). Instead, all transitions were directed downstream from a top state (state A) with the fastest growth rate (Fig 4F), toward three sink states (states C, D, and E), partly via an intermediate state (state B). The final state (state F) initially appeared disconnected from the network, but when the threshold for state transition occurrences in the network was decreased to 0.001 (corresponding to 1 transition in 1,000 days) rare transitions toward state C emerged.

To elucidate the correspondence between states from the different cell lines, we assigned the U3071MG and U3017MG cells to the U3065MG states via ssGSEA. While not all states corresponded in a 1-to-1 manner between cell lines (Fig 4G and H), we saw a clear correspondence between the fastest growing states (state 2 in U3065MG cells, state II in U3071MG, and state A of U3017MG). In all cases, GSEA results showed enrichment of cell cycle-related gene sets as well as neural progenitor-associated signatures (Garofano PPR, Zhong NPCs; Fig 5A). Similarly, the top states of U3065MG and U3071MG had clear similarities in terms of enrichment of mesenchymal signatures described by Neftel *et al* (2019) and Wang *et al* (2018). Thus, both the mesenchymal GBM cell lines appear to be driven by a mesenchymal state at top of the STAG network, the counterpart of which was not found in the classical cell line.

When relating the state signatures to TCGA survival, we noted that U3071MG-state I had a significant association to poorer survival (Cox coefficient = 1.230, $P < 0.01$), while the same was not true for the top state in U3017MG (Fig EV3). Taken together, this indicates that the top mesenchymal state in the two mesenchymal GBM cell lines predicts survival in TCGA patients, but that a similar trend could not be found for the rapidly proliferating, progenitor top state in the classical GBM cell line.

## GBM top states are transcriptionally similar to early embryonal neural precursor cells

To explore the relationship between our STAG networks of GBM and a normal neural developmental hierarchy, we compared our state signatures with embryonic scRNA-seq-data from a recent atlas of the embryonic human brain (Eze *et al*, 2021). In this atlas, embryonic brain cells from different Carnegie stages (12–22) are divided into 61 different clusters. We calculated the statistical overlap between each such embryonic cluster with our state signatures and sorted the embryonic clusters based on the mean Carnegie time of all cells in that cluster (Fig 5B). Interestingly, the top states in each STAG network (U3065MG-states 1 and 2, U3017MG-states

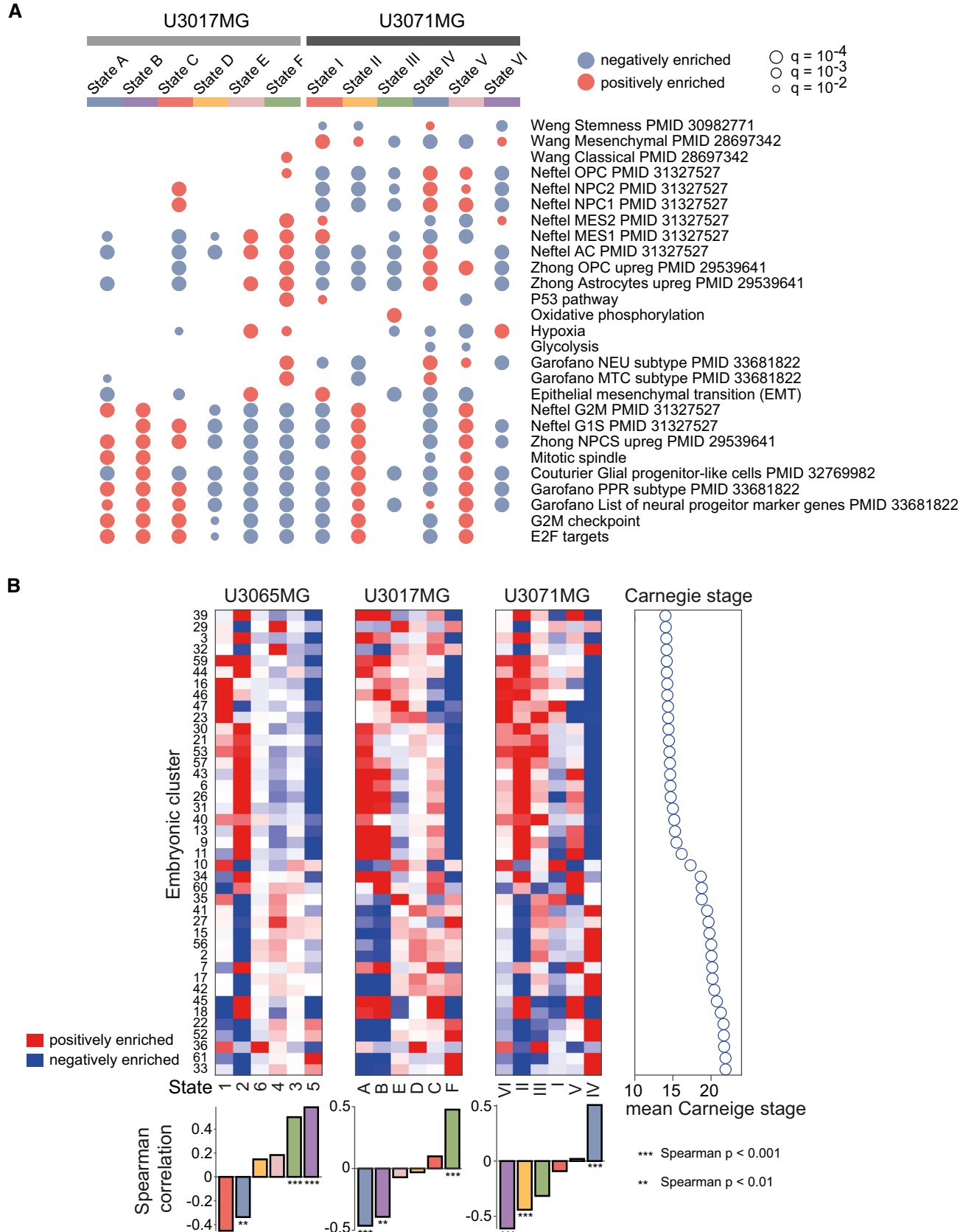

Figure 5.

◄

**Figure 5. Characterization of the states in the mesenchymal U3071MG and the classical U3017MG GBM cell lines.**

A  GSEA of the gene signatures for each of the six states in U3017MG and U3071MG. Blue and red indicate negative and positive enrichments of the pathway. Size of the dot indicates significance (proportional to the GSEA FDR *q*-value). Gene sets without PMID reference were obtained from the MSigDB Hallmarks database of gene sets. Abbreviations: oligodendrocyte progenitor cell (OPC), neural progenitor (NPC), mesenchymal (MES), astrocyte (AC), neuronal (NEU), mitochondrial (MTC), proliferative/progenitor (PPR).

B  State signatures for U3065MG (1–6), U3017MG (A–F), and U3071MG (I–VI) related to embryonic cluster definitions from Eze *et al*. The heatmap shows enrichment results; red and blue indicate positive and negative enrichments, respectively. The dot plot in the right panel shows mean Carnegie stage for cells in each embryonic cluster. Barplots below the heatmap show Spearman correlation ($n = 61$) between the enrichment profile for each state and the Carnegie time for each embryonic cluster. Spearman *P*-values are shown below barplots.

Source data are available online for this figure.

A and B, and U3071MG-state II) overlapped with cells from the earlier time points in the embryonic brain atlas, while downstream states showed a higher correlation with later time points. These results indicate that the GBM cell states and state transitions partly recapitulate the differentiation processes in the human developing brain.

## Therapeutic interventions in the state transition and growth model

Next, we aimed to explore how STAG can detect changes in cell state transitions during treatment. For this, we carried out a barcoding experiment in U3065MG cells, in which cells were exposed to two therapeutic agents (Fig 6A). Temozolomide (TMZ) was chosen as the standard-of-care drug and added at a concentration found in the cerebrospinal fluid of GBM patients (1 μg/ml), to reflect a clinically relevant dose (Ostermann *et al*, 2004). The cytokine BMP4 was chosen for its inhibitory effect on GBM growth (Piccirillo *et al*, 2006; Dalmo *et al*, 2020) and was administered at a dosage (1 ng/ml) shown to elicit a receptor response (Heemskerk *et al*, 2019) and to have an effect on cell fate choice (Lim *et al*, 2000), while not causing a complete stop of cell proliferation.

Following barcoded single-cell RNA profiling, we assigned the previously defined states 1–6 to each of the cells and fitted a more general version of the STAG model (Box 1), to obtain three STAG networks ($A_{untreated}$, $A_{TMZ}$, and $A_{BMP4}$) for U3065MG cells under each condition. Comparing the three models, we found that treatments had distinct effects on the growth rates of each cell state (Fig 6B). For instance, while TMZ was more effective in suppressing the proliferative state 2, BMP4 selectively suppressed the growth of state 5. Interestingly, both treatments led to a modest increase in the growth of state 1. Both treatments altered the state transitions in U3065MG cells. In BMP4-treated cells, we saw a selective decrease in the transitions from 2 to 4 (Fig 6C and D), whereas both treatments appeared to increase the rate of transitions from the neural progenitor-like state 2 to the OPC-like state 5 (Fig 6D and E).

The observation that treatments cause detectable changes in state growth rates and cell state transitions led us to ask whether the STAG model can be applied to long-term projections of the cell populations, both in terms of population size and in terms of the relative composition of cell states. Further mathematical analysis of the STAG model (Materials and Methods) showed that such predictions can be obtained from the eigendecomposition of the STAG network (*A* matrix). This standard type of mathematical matrix decomposition results in eigenvalues and eigenvectors, both of which have interesting interpretations in the context of our model (Fig 6E). First, the eigenvalues tell us how the population size will

develop. Specifically, from the theory of linear dynamical systems (Varfolomeev & Lukovenkov, 2018) the position of the eigenvalues of *A* along the real axis (in the complex number plane) tell us if the net number of cells will shrink toward zero (all eigenvalues negative), or grow exponentially (one or more positive eigenvalue). As a concrete example, TMZ-treated U3065MG cells, exposed to a dose of 1 μg/ml, had a positive growth rate, reflected by two positive eigenvalues (Fig 6E). We could thereby ask whether a small set of changes to transition rates might render the eigenvalues negative, thereby implying long-term eradication of the tumor cells. To test this idea, we adapted STAG to search for such modifications (Stability analysis, Materials and Methods), and found reduction in transitions from states 1, 2, 3, and 4 to state 5 would be required (Fig 6F). The second use of the eigendecomposition is that if the A-matrix has a real-valued eigenvector, this will predict the future cell state equilibrium (Steady-state distribution, Materials and Methods). By applying this to our STAG networks, we can predict the cell state equilibrium for cells kept untreated (Figs 6G and EV4), as well as for U3065MG cells during treatment with BMP4 and TMZ (Fig 6G).

We conclude that the STAG procedure can detect treatment-specific changes in state growth rates and state transitions. The eigendecomposition of the *A* matrix reflects a long-term projection of the system, in terms of net growth and state composition.

## Discussion

In this work, we have described a new strategy to assess the time-dependent heterogeneity of malignant tumors. The proposed STAG model quantitatively describes state transitions and growth, on a well-defined time scale. It can also characterize the effect of therapies and predict changes that would reduce tumor growth. Thereby, the method meets a need for principled and quantitative strategies to discuss the time-dependent heterogeneity of malignant tumors.

Applying STAG to three cases of GBM, we observed a hierarchically structured pattern of state transitions. The two mesenchymal subtype GBM cell lines (U3065MG and U3071MG) each had a mesenchymal top state which was followed by a downstream, rapidly proliferating state with a neural progenitor-like profile. In contrast, such a mesenchymal top state was not found in classical subtype U3017MG cells, where the rapidly proliferating state was found directly at the top. In the two mesenchymal lines, we found evidence of multi-directional switching, which was not observed in U3017MG cells, which were strictly hierarchical. All STAG networks contained sink states with no outgoing transitions at the bottom of the hierarchy. When unperturbed, our STAG models resemble the

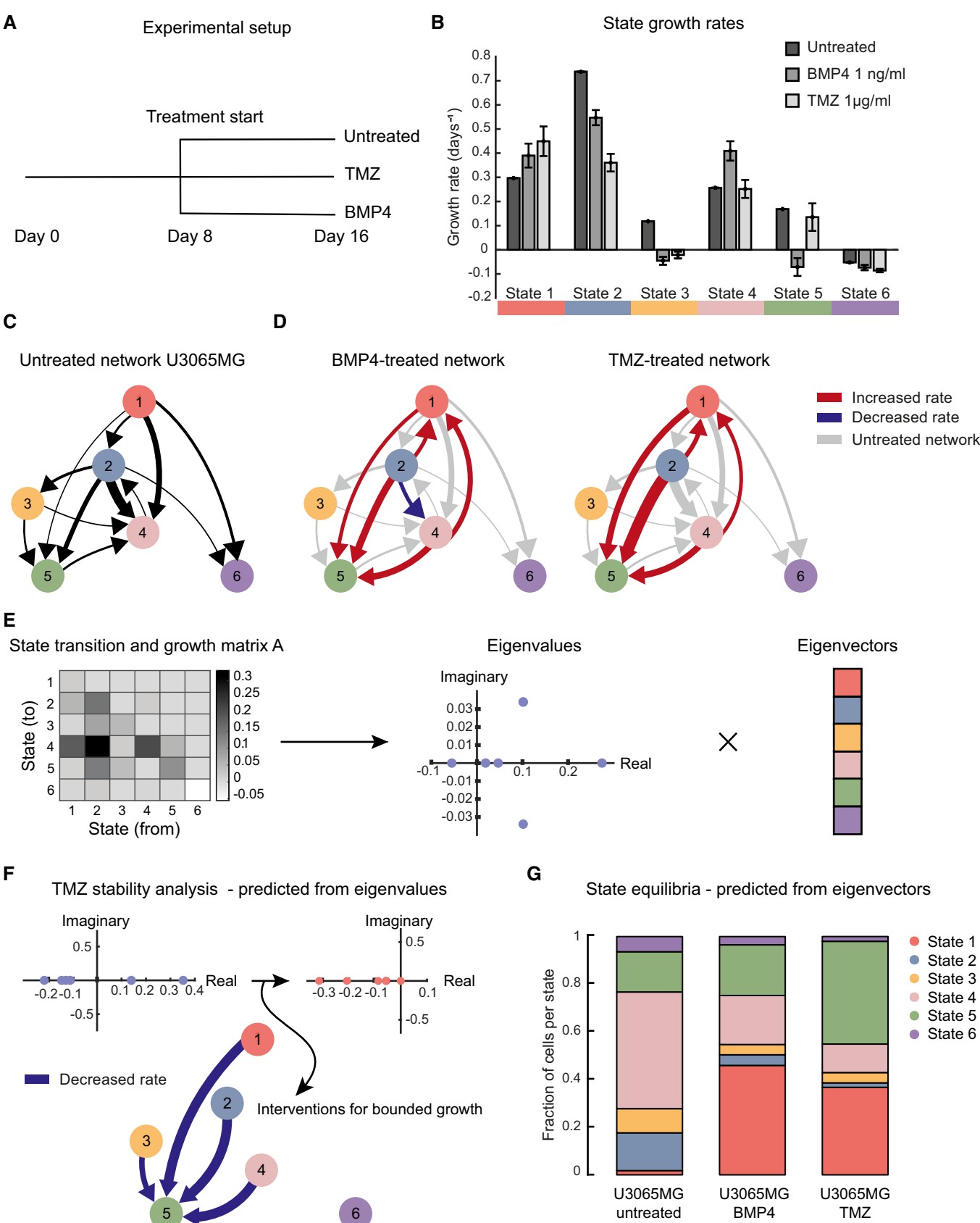

Figure 6.

◀

**Figure 6.  Employing the transition networks for assessing and predicting therapeutic interventions.**

A   Experimental design for U3065MG treatment experiments.
B   STAG estimates of the state-specific growth rates for untreated U3065MG cells, and cells treated with BMP4 and TMZ. Error bars indicate the 90% confidence interval, based on 1,000 bootstrap replicates.
C   Estimated network of state transitions in the untreated U3065MG cells. The thickness of the arrows correlates with the rate of transition (number of transitions per day).
D   Estimated network of state transitions for BMP4- and TMZ-treated U3065MG cells. Red indicates increased rate of transitions compared with untreated cells. Blue indicates decreased rate of transitions compared with untreated cells. Gray indicates transitions of the untreated network.
E   Conceptual image of how the STAG-derived A-matrix can be expressed as eigenvalues and eigenvectors, interpreted as measures of population stability and state equilibria, respectively.
F   Eigenvalues of the state transition network of TMZ-treated U3065MG cells and the predicted minimal intervention to obtain bounded growth in each case.
G   Predicted state equilibria for U3065MG cells in untreated condition and treated with BMP4 or TMZ.

proliferative hierarchy described by Lan *et al* (2017) where they use DNA barcoding in an *in vivo* setting. Similar to our findings, they see a gradual loss of barcodes over time. This can to a certain extent be explained by the sampling strategy, where all cells representing one barcode can be harvested at one time point by chance, and in part by the hierarchical structure, where barcodes that tagged cells at the bottom of the hierarchy at the experimental starting point are expected to disappear. Our detected states are enriched for markers found in previous studies (Zhong *et al*, 2018; Neftel *et al*, 2019; Couturier *et al*, 2020), and we also find a selective match between top states and early embryonic time points. This, together with an association between top states and survival, lends credence to the biological relevance of the detected state signatures.

Since microenvironmental factors are known to be important for GBM cell state choices (Calabrese *et al*, 2007; Wang *et al*, 2018; Neftel *et al*, 2019), future applications of this method should be considered to elucidate cell state transition organization when cells are exposed to environmental factors, such as hypoxia, or directly *in vivo*. In this work, we have chosen to interrogate the effects of the first-choice chemotherapeutic agent TMZ and BMP4, a physiological factor known to inhibit GBM cell proliferation. Both treatments changed the dynamics at the dose given, leading to altered rates of specific cell state transitions. In the particular case of TMZ, we saw an increased flux from the progenitor state 2, to state 5. State 5 is enriched for OPC signatures and mesenchymal transition signature, both of which are potentially associated with increased tumor cell motility (Kessaris *et al*, 2006; Iwadate, 2016). Using eigendecomposition, the STAG model could predict long-term state composition. Such analyses suggested that blocking transitions to state 5 might potentiate TMZ therapy (Fig 6F and G). An interesting future direction is therefore to identify such a blockade against state 5 transitions. Of note, our results do not imply that depletion of the top state would exhaust the GBM culture. They rather point toward a preferred cellular state organization in the unperturbed *in vitro* milieu with a large potential of adaptive plasticity. We acknowledge that the results may partly depend on the dose of each treatment. Here, we preferred to use doses that reflect a likely *in vivo* dose in patients (TMZ) and a low range dose where physiological effects have been observed (BMP4). In principle, the STAG framework can be adapted to make the changes in the STAG network dose-dependent, which may be an interesting avenue for future work.

When constructing the STAG model, we took a reductionistic approach and used a simplified system of 2D cultures of glioblastoma cells. The benefit of this approach is that a sufficient number of cells can be randomly sampled to support the estimation of cell state transitions. To the extent that state transitions and growth are modulated by 3D growth, or invasion of, e.g., blood vessels, immune cells, or other stromal components, these are obviously not captured in a 2D cell culture. It is notable, though, that our detected cell states overlap with gene signatures defined in clinical materials, embryonal brain, and other systems (Figs 3 and 5). An in-depth discussion of cell number and the benefits and disadvantages with 2D cultures is included in the Appendix. The application of STAG to the 3D culture setting, with appropriate computational extensions, is reserved for future work.

In the presented strategy, single-cell lineage tracing is implemented in a pooled setting from which cells are sampled at random. The two benefits of this are that it permits a relatively unbiased detection of cell states present in the culture, as well as separate estimates of transitions and growth parameters of each state. Subcloning schemes, by comparison, will focus predominantly on the subset of cells that can clonally expand in solitude (14–49%, Segerman *et al*). It is, however, important to apply STAG with an awareness of technical factors and choices that can potentially influence the results. Such factors include the experimental design (see Appendix) as well as the processing of the single-cell data, and the definition of cell states. For instance, when processing the RNA data, we here use cell cycle normalization based on the method by Butler *et al* (2018), designed to remove differences between cells in various cell cycle phases while maintaining the difference between cycling and non-cycling cells. In the context of our data, this method conserved phenotypically relevant signals (such as the up-regulation of cell cycle genes in fast-growing state 2 in U3065MG cells, c.f. Fig 3). In contexts where distinctions between cell cycles phase are essential, we would recommend using STAG without cell cycle correction. To identify states, we have used unsupervised clustering, commonly used within the GBM field (e.g., Phillips *et al*, 2006; Verhaak *et al*, 2010; Neftel *et al*, 2019). In principle, STAG could also be used with external state definitions, using ssGSEA to assign each cell to a state based on other predefined signatures. Last, an important analytical choice is the number of states. Here, the number of states was chosen as 6, in each cell line analyzed, based on data-driven considerations (Appendix Figs S3, S5, and S6). We find that network structures are overall kept intact with variations in cluster number, e.g., 5 or 7 clusters (Figs EV2 and EV5) and can detect biologically relevant signatures representing each state, substantiating the validity of our approach.

Looking ahead, we see several potential applications and generalizations of STAG for cancer studies. In GBM, applying our procedure across a broader set of patient samples and drugs would help elucidate how cell state transitions differ across patient subgroups and also explore if and how different classes of targeted drugs change state transition rates. Potentially, such an extended study may open for targeted interventions aimed at re-distributing cell states (c.f. Fig 6F and G). Extending the STAG algorithm itself, important directions include the investigation of non-linear versions of the model, and extending the model to capture, e.g., drug dose-dependent changes in cell STAG rates. The STAG software is freely available at GitHub. The provided modeling framework is not limited to glioma models; it can be applied to data sets generated from any cancer model using the described lentiviral barcoding strategy. STAG is designed to run with any number of time points (2 or more). Considering that tumor plasticity is a widely observed phenomenon in experimental oncology, we foresee that STAG will be applied to other forms of cancer than GBM as a tool to resolve and predict cell state dynamics.

# Materials and Methods

## Experimental methods

### Perturb-seq library preparation
The Perturb-seq GBC library was a gift from Jonathan Weissman (Addgene ID #85968) (Adamson et al, 2016). The library was expanded in NEB® 5-alpha electrocompetent E. coli (New England Biolabs, Ipswich, MA) according to the manufacturer's protocol, ensuring at least a 100× coverage to maintain library diversity. To produce lentivirus, library plasmids were co-transfected with plasmids pLP1, pLP2, and VSVg (5 μg/ml of each, Thermo Fisher Scientific, Boston, MA) using PEI (100 μg/ml, PolySciences, Warrington, PA) into HEK293T packaging cells. Virus supernatant was collected at 48 and 72 h after transfection, and virus was purified by ultracentrifugation at 72,000 g for 2 h, for 2 h, re-suspended in DMEM/F12 (Thermo Fisher Scientific), and stored in aliquots at −80°C until use.

### Human glioblastoma cell culture
U3017MG, U3065MG, and U3071MG cells were obtained from the Human Glioblastoma Cell Culture (HGCC) Biobank (Xie et al, 2015) and cultured adherently on laminin (Sigma-Aldrich, St. Louis, MO)-coated plates in a 1:1 mix of Neurobasal and DMEM/F12 medium supplemented with B27 (without retinoic acid) and N2 (Thermo Fisher scientific) and human recombinant EGF and FGF2 (10 ng/ml, PeproTech, Rocky Hill, NJ). For cell detachment, StemPro Accutase (Thermo Fisher scientific) was used. Cell line authenticity was confirmed by STR profiling, and the cell lines displayed no mycoplasma contamination (Eurofins Genomics). Tumor sample collection was approved by the Uppsala regional ethical review board, number 2007/353; informed consent was obtained from all subjects included. Experiments conformed to the principles set out in the WMA Declaration of Helsinki and the Department of Health and Humans Services Belmont Report.

### Growth rate determination
To determine the growth rate at the cells exponential growth phase, non-barcoded cells were seeded in quadruplicate wells at five different starting densities ranging between 8 and 500 cells/mm$^2$. Cells were kept for up to 13 days with readouts of cell count at days 1, 3, 6, 8, 10, and 13 using Trypan blue exclusion on Countess Cell Counting Chamber Slides (Invitrogen) or alamarBlue Assay (Invitrogen). 95% confidence intervals were determined using GraphPad Prism 6.

### Barcoding procedure
Human glioblastoma cells between passages 6 and 12 were infected with the Perturb-seq lentiviral library. Functional virus titer was determined by measuring the percentage of BFP-positive (BFP$^+$) cells by flow cytometry for each cell line. To minimize potential off target effects of the lentiviral tagging and to ensure accurate estimates of cell state changes, a multiplicity of infection of < 0.1 was used, rendering single barcodes in more than 95% of barcoded cells (see Appendix for details). Briefly, cells were incubated with the lentivirus as a single-cell suspension in 20 μl of media for 4 h at 37°C and seeded onto a laminin-coated 60-mm dish. The next day, virus-containing media was replaced with fresh media. Cells were then allowed to propagate for 4–7 days. At experimental day 0, 2,500 BFP$^+$ cells were sorted into one well on a 96-well plate, giving an expected 96.7% of cells with a unique barcode combination, according to calculations based on (Lan et al, 2017) (see Appendix for details). To confirm this, the remaining unsorted cells were frozen down in media with 10% DMSO in −150°C and later subjected to single-cell RNA sequencing. To verify that barcoding itself having a strong effect on states, we compared barcoded vs non-barcoded cells and found no differentially expressed pathways (Appendix).

### Time series experiment
For U3065MG, barcoded cells were detached at experimental day 7 and 14. 80% of the cells frozen down, while the remaining 20% were seeded into a new well. After 21 days at the experimental endpoint, all cells were frozen down. For U3017MG and U3071MG, harvested cells were freshly prepared for single-cell RNA sequencing and samples were obtained for experimental days 7 and 14.

### Drug treatment experiment
At day 8, barcoded U3065MG cells were detached and 70% of the cells were frozen down. The remaining cells were equally divided into three new wells on a 96-well plate. When attached, cells were treated every second day with 1 ng/ml BMP4 (Thermo Fisher Scientific), 1 μg/ml Temozolomide (Sigma-Aldrich) or remained untreated. At day 16, cells from all wells were frozen down.

### FACS sorting and flow cytometry
Cells were detached using TrypLE Select (Thermo Fisher scientific), pelleted, and re-suspended in FACS buffer (0.5% BSA/2 mM EDTA in PBS), followed by staining with fluorophore-conjugated antibodies for 20 min, 4°C. Antibodies used were CD24-BV421 (BD PharMingen, #562789), CD44-FITC (BD PharMingen, #555478), and HLA-DR-APC-Vio770 (Miltenyi Biotec, #130-111-792). Isotype control antibodies for each fluorophore were from the same

companies as the primary antibodies. Cell sorting was performed on a BD FACSAria III Cell Sorter and flow cytometry on a BD LSR Fortessa instrument (BD Biosciences, for instrument settings, see Appendix Supplementary methods). Data analysis was performed using the Kaluza Analysis Software (Beckman Coulter). The optical configuration on the BD FACSAriaIII cell sorter and the BD LSR Fortessa flow cytometer is presented in Table S2.

### Single-cell RNA sequencing

Single-cell RNA sequencing libraries were prepared using the Chromium Single Cell 3′ Library Gel Bead Kit v3 according to the manufacturer's instructions (10× Genomics). Cryo-preserved cells were washed and re-suspended in 0.1% BSA in PBS immediately before loading on a Chromium Single Cell B Chip (10× Genomics), with a target capture of 10,000 cells. Libraries' quality was assessed using the Agilent BioAnalyzer High-Sensitivity DNA assay then sequenced using Illumina's NextSeq 500 platform.

## Computational methods

### Data processing

The data were processed before further analysis in R (R Core Team, 2019) using the package Seurat (Butler *et al*, 2018). Cells containing less than 500 genes and genes present in less than 3 cells were filtered out, and data were log normalized and row centered. To prevent the downstream analysis from only capturing cell cycle dynamics, the difference in cell cycle phase between cells was removed from the data by following the workflow proposed by Seurat (Butler *et al*, 2018). Each cell was assigned scores based on gene markers for the S- and G2/M-phases, and the difference between these scores was regressed out.

### Differential expression analysis between large and small clones

Cells were classified as small, intermediate, or large clones based on their associated barcodes. Small clones were defined as those containing a barcode present in ≤ 6 cells, large clones contained a barcode present in ≥ 17 cells, and intermediate clones contained barcodes present in cells in between these numbers. The cutoff values were chosen to obtain equally sized groups. A two-sample t-test was performed to identify genes that were differentially expressed between the large and small clones. GeneIDs and t-statistics were extracted for all genes with FDR < 0.25 and used to create a ranked list. This was subsequently submitted to GSEA in the pre-ranked mode using default parameters. In addition to the hallmark gene sets from the MSigDB collections, gene signatures defining stemness (Patel *et al*, 2014) and transcripts regulated by the transcription factor SOX2 (Singh *et al*, 2017) were included in the GSEA analysis.

### Deciding number of states

For determining the optimal number of clusters (*k*) in which to partition the data, consensus clustering and bootstrap network estimation were used. To start with, consensus clustering for *k* = 2–8 was done using the R-package ConsensusClusterPlus (Wilkerson & Hayes, 2010). 80% of the cells were sampled from the data set and clustered using *k*-means. This process was repeated 100 times. Appendix Fig S3A shows the generated consensus matrices for *k* = 2–8 and Appendix Fig S3B a plot

showing the relative change in the cumulative distribution function (CDF) curve.

Based on the consensus clustering results, bootstrap network estimation was carried out for *k* ≥ 4. Again, 80% of the cells were sampled and the transition network was estimated, this process was repeated 100 times for *k* = 4–8. Model robustness was scored by calculating the consistency of growth rate values between runs (Appendix Fig S3C).

For the additional cell lines U3017MG and U3071MG, cluster number was decided based on results from the consensus clustering performed using the R-package ConsensusClusterPlus (Wilkerson & Hayes, 2010), Appendix Figs S5 and S6.

### Definition and initial assignment of cells into transcriptional states

Cells were divided into states based on their gene expression using the *k*-means clustering algorithm in R. Marker genes for each state were defined through differential expression analysis in Seurat using the test MAST (Finak *et al*, 2015). GeneIDs and log2FC-values were extracted for all genes with adjusted *P*-value < 0.01 and used to create ranked gene lists for each state. These were subsequently submitted to GSEA in pre-ranked mode using default parameters. In addition to the hallmark gene sets from the MSigDB collections, gene signatures derived from various publications were used, and pmid is indicated in Fig 3A.

### State assignment using single-sample scoring

The R-package singscore (Foroutan *et al*, 2018) was used to score individual cells in the drug treatment experiment against gene signatures derived from the previously defined states. Cells were assigned to the state with the highest obtained score. For the integration of single-sample scoring with the established computational pipeline for analyzing barcoded data, we implemented the singscore algorithm in MATLAB.

### Time-dependent transition and growth model

At a given time point t, we model the population of GBM cells as the tuple $X(t) = \{X_1, X_2, ...X_k\}(t)$, where each $X_i(t)$ is the number of cells in transcriptional state *i* at time *t*. In a given time interval, $\delta t$, each individual cell can undergo three types of discrete events:

- Growth, $X_i \rightarrow X_i + 1$, with rate constant $\alpha_i$
- Death, $X_i \rightarrow X_i - 1$ with rate constant $\beta_i$
- Transition between states i and j, $(X_i, X_j) \rightarrow (X_i - 1, X_j + 1)$ with rate constant $\gamma_{ji}$

From a given starting state (e.g., 14, 0, and 4 cells in states 1, 2, and 3, respectively), this model can be forward-simulated using Gillespie's method (Gillespie, 1976) to yield a sample of how $X_1, X_2, ...$ evolve over time. Here, we are interested in the inverse problem, i.e., a situation where $X_1, X_2, ...$ are observed at several time points but where the parameters $(\alpha, \beta, \gamma)$ are unknown. One approach to estimate the parameters is to employ Markov chain Monte Carlo methods (MCMC). This, however, is computationally costly. For an efficient approximate estimation, we employ the following continuum approximation. If we view the vector $X$ as concentrations rather than counts, which is a good approximation for higher values of $X$, we can model the growth and transitions as

$$dX_i/dt = (\alpha_i - \beta_i)X_i - \sum_{k \neq i}\gamma_{kj}X_i + \sum_{j \neq i}\gamma_{ij}X_j, \qquad (6)$$

i.e., the rate of change in cells in state $i$ is a function of growth, death, and transitions to/from that state. This set of linear first-order differential equations can be expressed on a matrix format as follows:

$$\frac{d}{dt}X(t) = AX(t). \qquad (7)$$

The elements of $A = \{a_{ij}\}$ are $a_{ij} = \gamma_{ij}$ when $i \neq j$ and $a_{ij} = \alpha_i - \beta_i - \sum_{k \neq i}\gamma_{kj}$ when $i = j$. The solution is given by ($e$ denoting a matrix exponential):

$$X(t + \delta t) = e^{A\delta t}X(t). \qquad (8)$$

Thus, if we have an observation at two time points $X(t_1)$ and $X(t_2)$,

$$X(t_2) = e^{(t_2 - t_1)A}X(t_1)\frac{(1 - \eta_1)\eta_2}{\eta_1} \qquad (9)$$

where $\eta_s$ is the fraction of cells used for sequencing at each time point $t_s$, $s = 1, 2, 3$ (7, 14, and 21 days, respectively). Using a first degree approximation of the matrix exponential, $e^A t \approx I + At$, the sum-of-squares error over all barcodes is thus given by:

$$E(X, A) = \sum_{s=2,3}||X(t_s) - (I + A(t_s - t_{s-1})X(t_{s-1})\frac{(1 - \eta_{s-1})\eta_s}{\eta_{s-1}}||_{Fro} \qquad (10)$$

which is solved by optimization to yield an estimate of $A$, from which estimates of $\gamma_{ij}$ and the growth parameters $r_i = \alpha_i - \beta_i$ are obtained. Important to note here is that the linear approximation of the system allows us to formulate this as a regression problem, which has a unique solution as $X$ is full rank.

The specific optimization problem solved is as follows:

$$\min_{A} \quad E(X, A)$$
$$\text{s.t.} \quad \sum_{i \neq j}|a_{ij}| \leq \lambda \ . \qquad (11)$$
$$a_{ij} \geq 0, \ i \neq j$$

Note that non-diagonal elements of $A = \{a_{ij}\}$ are constrained to non-negative values (because they are rates) and that the sum of absolute values is constrained to ensure sparsity of the solution (a so called l1 or lasso penalty). The solution is obtained in MATLAB using the CVX solver (cvxr.com).

### Effects of treatments on network parameters

We represent treatments as perturbations of the model parameters, i.e.,

$$A = A_{\text{joint}} + \Delta A_{\text{treatment}} \qquad (12)$$

Accordingly, we can define a global error across all treatments as

$$E_{\text{global}}(X, A, \Delta A_1, \Delta A_2, ...) = \sum_{t \in \text{ treatments}} E(X_t, A + \Delta A_t) \qquad (13)$$

The associated optimization problem is.

$$\min_{A, \Delta A_1, \Delta A_2, ...} \quad E_{\text{global}}(X, A, \Delta A_1, \Delta A_2, ...)$$
$$\text{s.t.} \qquad ||A^{\text{nd}}||_1 \leq \lambda_1$$
$$||A^{\text{nd}} + \Delta A_i^{\text{nd}}||_1 \leq \lambda_2, \ i = 1, 2, ...$$
$$A^{\text{nd}} \geq 0 \qquad (14)$$
$$A^{\text{nd}} + \Delta A_i^{\text{nd}} \geq 0, \ i = 1, 2, ...$$
$$||A - A_{pri}||_{Fro} \leq \Omega$$

where notation $A^{\text{nd}}$ denote matrix $A$ with diagonal elements set to zero, i.e., the non-diagonal part of $A$. The first two constrains impose sparsity on the joint network and the treatment-specific networks, respectively. The following two constraints ensure non-negativity of the rates in the joint and treatment-specific networks, respectively. The last constraint is the tolerated deviation of $A$ from a prior transition matrix. This is an optional constraint, and the deviation is defined by the user. The solution is obtained in MATLAB using the CVX solver (cvxr.com).

### Cross-validation for tuning lasso penalties

To decide the lasso penalty for the baseline network, the original (untreated) data set was split in two, one training and one test set. The matrix $A$ was solved for the training set for values of the lasso penalty from 0.5 to 2 and for each value, and the distance to the test set (error) was calculated. The same procedure was done to decide the second penalty but using the data from the treatment experiment (Appendix Fig S7).

### Benchmarking the transition and growth model

A large number of simulations were performed to assess the validity of our model in the following manner.

The Gillespie algorithm (Gillespie, 1976) was used to simulate synthetic data sets using user-defined transition and growth rates. These were random but chosen to generate a mean rate in the matrix to around 0.03 days$^{-1}$. Our model was fitted to the simulated data set and all transition and growth rates were estimated, emulating the workflow for real data sets. The distance between the input and output transition matrix (calculated as the Frobenius norm) was calculated and used as a goodness-of-fit measure. The simulations were done for several varying conditions, number of states from 2 to 10 and for each number of state from one transition to a fully connected network. For each connectivity value, the sampling interval was varied from 1 to 10 days. However, the allowed number of sampling times was kept fixed to three times and the sampled fraction to 80% to ensure comparability between runs.

The results are visualized in Appendix Fig S2, showing one plot for each number of states. The x-axis displays an increasing value for the days between sampling, and the y-axis displays an increasing value for the connectivity (number of links in the network). Values plotted are the distance between input and output networks, described above. From the plots, we see that for some sets of conditions the model re-creates the transition matrix with high accuracy (blue areas, low error), while other sets of conditions have a higher estimation error (yellow areas, high error), e.g., in the cases where we have many states and sampling is done at a too high or too low frequency.

### Bootstrapping

Bootstrapping was used to estimate confidence intervals for growth rates derived from the STAG model (Figs 2F and 6B). A random data set of the same size as the original data set ($X(t)$, described above) was constructed by sampling with replacement from the original data set. The growth rates were estimated using the STAG model. The process was repeated 1,000 times, and the confidence intervals were calculated based on the bootstrap outputs.

### Galton–Watson model

To relate the per-barcode cell counts in Fig 1C to a Galton–Watson process, we carried out a simulation, in which we started with a hypothetical population of cells, each carrying 1 barcode (Fig 1D, day 0). We subsequently simulated a 1-state version of our model (with growth rate $\alpha = 0.2$/day and death rate $\beta = 0.03$/day, no transitions) and used Gillespie's algorithm to simulate a distribution of cells per barcode (Fig 1D, day 7–day 21). The simulation used a number of barcode (1,217) and assumed that 80% of cells were harvested for sequencing each week, as in our experimental protocol.

### Validation in external data sets

We retrieved transcriptomic profiles and matched survival data for 501 GBM patients from the TCGA cohort (Brennan *et al*, 2013). Each patient in the data set was scored against our derived state signatures using the ssGSEA method from the R-package gsva (Hänzelmann *et al*, 2013). Cox's proportional hazards models were generated using the enrichment score of each state as independent predictor, with clinical covariates (age at diagnosis and gender) included. To relate our cell state signatures to embryonic human brain signatures, we used the data from Eze *et al* (2021), obtained from (https://cells-test.gi.ucsc.edu/?ds=early-brain). We used the 61 clusters provided by the authors. The mean Carnegie age of each cluster (Fig 5B) was calculated as the average Carnegie stage of all cells belonging to that cluster. We computed the overlap between the embryonic clusters (genes with log fold change > 0.25 in each respective embryonic clusters 1–61) and our state signatures (genes with log fold change > 0.25 in each of our cell state signatures) by Fisher's exact test. When the gene set overlap was higher than expected, the enrichment score was $-\log(P)$ (i.e., a positive number), where $P$ is the Fisher $P$-value (red in Fig 5B). Conversely, when the gene set overlap was smaller than expected, the enrichment score was defined as $+\log(P)$ (i.e., a negative number.). We thus obtained a profile for each state in terms of enrichment scores, which was correlated with mean Carnegie time (Spearman correlations, Fig 5B).

### Stability analysis

To predict the minimal intervention $\Delta A$, we solved the problem:

$$\min_{\Delta A} \quad \|\Delta A_1\| \tag{15a}$$

$$\text{subject to} \quad A + \Delta A \text{ is stable.} \tag{15b}$$

To solve this problem, we used a convex stability criterion (Box 1, equation 5), based on stability (Zavlanos *et al*, 2011).

### Steady-state distribution

Consider a short time interval between $t$ and $t + dt$. Then, $y(t + dt) = y(t) + y'(t)dt + \varepsilon$ (where the residual $\varepsilon$ can be made arbitrarily small by reducing $dt$). Since $y'(t) = Ay(t)$, we have the following:

$$y(t + dt) = (I + Adt)y(t) + \varepsilon. \tag{16}$$

If the relative proportions of the states are at equilibrium at time $t$, this is the same as saying that $y(t + dt) = ky(t)$, i.e., $y(t)$ merely scales by a constant $k$.

$$ky(t) = (I + dtA)y(t) \tag{17}$$

Call the solution to this equation $v = y(t)$. If we apply the substitution $k = 1 + \lambda dt$ (where $\lambda$ is some constant), we obtain.

$$(I + \lambda dt)v = (I + Adt)v \tag{18}$$

which simplifies to the eigenvector equation

$$\lambda v = Av. \tag{19}$$

In other words, even when the cell population grows, there can be a steady state of the relative proportions of the states. When such a steady state exists, the proportion $f_i$ of each state $i = 1, 2...$ is given by.

$$f_i = \frac{v_i}{\sum(v_i)}. \tag{20}$$

where $v$ is a real, non-negative eigenvector of $A$. The existence and uniqueness of such a steady state of proportions, for a given matrix $A$, is a nuanced question. Generally, if all states are connected and all have positive growth (ie all entries of $A$ are positive), the Perron–Frobenius theorem says that a real-valued, positive eigenvector $v$ exists. The solution is not guaranteed to be unique. For instance, for an identity matrix (no transitions) any state composition is a proportional steady state.

## Data availability

The data sets and computer code produced in this study are available in the following database

- scRNA-seq data: ArrayExpress E-MTAB-9296 (https://www.ebi.ac.uk/arrayexpress/experiments/E-MTAB-9296/) and E-MTAB-10615 (https://www.ebi.ac.uk/arrayexpress/experiments/E-MTAB-10615/).
- STAG model: Github (https://github.com/idalarsson/STAG)
- Code used to conduct analyses:

  https://figshare.com/projects/Modeling_glioblastoma_heterogeneity_as_a_dynamic_network_of_cell_states/113166

**Expanded View** for this article is available online.

## Acknowledgements

We thank the Swedish Cancer Society, Swedish Childhood Cancer Foundation, Swedish Research Council, and the Swedish Strategic Research Foundation for financial support. Open access funding was provided by Uppsala University. Sequencing was performed by the SNP SEQ Technology Platform in Uppsala. The facility is part of the National Genomics Infrastructure (NGI) Sweden and Science for Life Laboratory. We also thank Dirk Pacholsky at the BioVis platform, Uppsala University, for help with FACS sorting.

## Author contributions

ED designed and conducted the lineage-tracing experiments. RE and MD performed the scRNA-seq, and RE performed the first pre-processing of the data. IL and SN developed the STAG model, and IL performed the computational analyses with support from SN and RJ. MN performed the FACS and flow cytometry experiments. MN and AS characterized the U3065MG clone. IL, ED, BW, and SN wrote the first version of the paper, and all authors assisted in editing the paper. BW and SN guided the study.

## Conflict of interest

The authors declare no conflict of interest. RE is presently an AstraZeneca employee.

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
