## [Review Process File · Molecular Systems Biology]

Modeling glioblastoma heterogeneity as a dynamic network of cell states

Ida Larsson, Erika Dalmo, Ramy Elgendy, Mia Niklasson, Milena Doroszko, Anna Segerman, Rebecka Jörnsten, Bengt Westermark, and Sven Nelander

DOI: [10.15252/msb.202010105](https://doi.org/10.15252/msb.202010105)

Corresponding author: Sven Nelander (sven.nelander@igp.uu.se)

Review Timeline:

Submission Date:	13th Nov 20
Editorial Decision:	17th Nov 20
Appeal Received:	24th Nov 20
Editorial Decision:	26th Jan 21
Revision Received:	1st Jul 21
Editorial Decision:	4th Aug 21
Revision Received:	16th Aug 21
Accepted:	17th Aug 21

Editor: Maria Polychronidou

Transaction Report:

Manuscript Number: MSB-2020-10105

Thank you for submitting your manuscript "Modeling glioblastoma heterogeneity as a network of cell states" to Molecular Systems Biology.

We have now considered your manuscript and I regret to inform you that we have decided to not send it out for peer review.

In this study, you introduce inheritable barcodes into a culture of patient-derived glioblastoma (GBM) cells, propagate the cells in vitro and profile the transcriptome of cell fractions sampled at regular time intervals. You then use the State Transition and Growth model to estimate the main transcriptional states, their growth rates and state transitions. We appreciate that you observe the presence of a six-state hierarchical organization. We also acknowledge that you examine the effect of BMP and temozolomide and report that they have different effects on state transitions. However, we feel that as it stands the study does not seem to provide a sufficiently in depth understanding of the six states, their transitions, their role in GBM progression and the implications for therapeutic interventions. Overall, we are not convinced that the study provides the degree of systems-level biological insight and the kind of decisive methodological advance with a demonstrated potential to aid new biological discoveries that would be required for publication in Molecular Systems Biology.

I am very sorry to have to disappoint you on this occasion, but I hope that this early decision will allow you to decide how to proceed with your manuscript without undue delay.

Thank you for the careful assessment of our paper entitled 'Modelling glioblastoma heterogeneity as a network of cell states' by Ida Larsson, Erika Dalmo et al (MSB-2020-10105). We understand that MSB receives a high number of high-quality submissions, and look forward to submitting our work to MSB in the future. In this instance, we are concerned that MSB Editors seem to have underappreciated the conceptual novelty of the paper. In support of a continued review process, we would like to argue the following scientific points:

1. Probing change in RNA signatures at the single-cell level is likely to become one of the central developments in the single-cell field in the coming few years. Current long-term RNA trajectory estimators are mainly focused on deducing developmental hierarchies in healthy tissue [1-4] and thus are not optimized to find bidirectionality in cell fate choices, nor accounting for cell growth.

By contrast, ours is a first well-defined method to simultaneously track the cellular state changes in combination with high-resolution single-cell lineage tracing in a growing tumor cell population.

2. On the same account, probing therapeutically induced changes of RNA trajectories will likely be a top priority in cancer pharmaco-genomics in the coming few years. Our report presents a first clear mathematical description of how this can be done, by solving for the network differential between treated and untreated cells. As demonstrated for TMZ and BMP4 treatment, we present a new principle for drug assessment; instead of focusing on e.g. growth or differentiation, we simply quantify drug action as the joint re-wiring of RNA trajectories and growth parameters.

Being able to probe these co-dependent processes has high principal interest for cancer drug development.

3. In brain tumor research (a vast field in itself), there is an ongoing debate whether cancer stem cells should be the primary cell population for therapeutic targeting [5,6] or if such (stem cell-centered) efforts are futile due to extensive tumor cell plasticity [7,8]. Our report does add a new type of orthogonal evidence to support the existence of a hierarchical structure in glioblastoma. Crucially, the identified organization is transformed into a bidirectional one upon exposure of one of the current standard of care treatments for patients as well as for a proposed differentiation therapy agent under clinical trial [9]. We emphasize this point since the comments from the editor read as if we were mainly concerned with introducing a new framework for classification of malignant glioblastoma cells; our intent is rather to introduce a broadly applicable concept for deducing cellular state organizations in patient-derived solid tumor cells, with glioblastoma as a case-in-point.

4. Last, our paper presents a substantial advance in model-based data interpretation. A new mathematical formulation is presented (STAG) that accounts for the growth and transitions of barcoded tumor cells. An algorithm for solving the STAG model for barcoded data is presented, which enables the estimation of transition structure and growth under a range of conditions. Furthermore, a novel eigenvalue criterion is proposed, which can be used to estimate optimal re-wiring of the tumor cells to induce stable or shrinking net growth. We point this out since the comments from the Editor read as if STAG were a pre-existing concept, which would be a misunderstanding. STAG has many potential applications to the investigation of patient-derived cell models, e.g. to support time-dependent modeling of responses to therapeutic interventions in vitro or in organoid systems.

With these clarifications, we ask that the paper is re-assessed for peer review. We will be happy to supplement our submission with data and explanations to further underline the above points. We look forward to your response,

REFERENCES

- 1 Raj B, Wagner DE, McKenna A, Pandey S, Klein AM, Shendure J, Gagnon JA, Schier AF. Simultaneous single-cell profiling of lineages and cell types in the vertebrate brain. *Nat Biotechnol.* 2018 Jun;36(5):442-450. doi: 10.1038/nbt.4103. Epub 2018 Mar 28. PMID: 29608178; PMCID: PMC5938111.
- 2 Spanjaard B, Hu B, Mitic N, Olivares-Chauvet P, Janjuha S, Ninov N, Junker JP. Simultaneous lineage tracing and cell-type identification using CRISPR-Cas9-induced genetic scars. *Nat Biotechnol.* 2018 Jun;36(5):469-473. doi: 10.1038/nbt.4124. Epub 2018 Apr 9. PMID: 29644996; PMCID: PMC5942543.
- 3 Weinreb C, Rodriguez-Fraticelli A, Camargo FD, Klein AM. Lineage tracing on transcriptional landscapes links state to fate during differentiation. *Science.* 2020 Feb 14;367(6479):eaaw3381. doi: 10.1126/science.aaw3381. Epub 2020 Jan 23. PMID: 31974159; PMCID: PMC7608074.

4 Bidy BA, Kong W, Kamimoto K, Guo C, Waye SE, Sun T, Morris SA. Single-cell mapping of lineage and identity in direct reprogramming. *Nature*. 2018 Dec;564(7735):219-224. doi: 10.1038/s41586-018-0744-4. Epub 2018 Dec 5. PMID: 30518857; PMCID: PMC6635140.

5 Couturier CP, Ayyadhury S, Le PU, Nadaf J, Monlong J, Riva G, Allache R, Baig S, Yan X, Bourgey M, Lee C, Wang YCD, Wee Yong V, Guiot MC, Najafabadi H, Mistic B, Antel J, Bourque G, Ragoussis J, Petrecca K. Single-cell RNA-seq reveals that glioblastoma recapitulates a normal neurodevelopmental hierarchy. *Nat Commun*. 2020 Jul 8;11(1):3406. doi: 10.1038/s41467-020-17186-5. Erratum in: *Nat Commun*. 2020 Aug 7;11(1):4041. PMID: 32641768; PMCID: PMC7343844.

6 Lan X, Jörg DJ, Cavalli FMG, Richards LM, Nguyen LV, Vanner RJ, Guilhamon P, Lee L, Kushida MM, Pellacani D, Park NI, Coutinho FJ, Whetstone H, Selvadurai HJ, Che C, Luu B, Carles A, Moksa M, Rastegar N, Head R, Dolma S, Prinos P, Cusimano MD, Das S, Bernstein M, Arrowsmith CH, Mungall AJ, Moore RA, Ma Y, Gallo M, Lupien M, Pugh TJ, Taylor MD, Hirst M, Eaves CJ, Simons BD, Dirks PB. Fate mapping of human glioblastoma reveals an invariant stem cell hierarchy. *Nature*. 2017 Sep 14;549(7671):227-232. doi: 10.1038/nature23666. Epub 2017 Aug 30. PMID: 28854171; PMCID: PMC5608080.

7 Dirkse A, Golebiewska A, Buder T, Nazarov PV, Muller A, Poovathingal S, Brons NHC, Leite S, Sauvageot N, Sarkisjan D, Seyfrid M, Fritah S, Stieber D, Michelucci A, Hertel F, Herold-Mende C, Azuaje F, Skupin A, Bjerkvig R, Deutsch A, Voss-Böhme A, Niclou SP. Stem cell-associated heterogeneity in Glioblastoma results from intrinsic tumor plasticity shaped by the microenvironment. *Nat Commun*. 2019 Apr 16;10(1):1787. doi: 10.1038/s41467-019-09853-z. PMID: 30992437; PMCID: PMC6467886.

8 Neftel C, Laffy J, Filbin MG, Hara T, Shore ME, Rahme GJ, Richman AR, Silverbush D, Shaw ML, Hebert CM, Dewitt J, Gritsch S, Perez EM, Gonzalez Castro LN, Lan X, Druck N, Rodman C, Dionne D, Kaplan A, Bertalan MS, Small J, Pelton K, Becker S, Bonal D, Nguyen QD, Servis RL, Fung JM, Mylvaganam R, Mayr L, Gojo J, Haberler C, Geyeregger R, Czech T, Slavc I, Nahed BV, Curry WT, Carter BS, Wakimoto H, Brastianos PK, Batchelor TT, Stemmer-Rachamimov A, Martinez-Lage M, Frosch MP, Stamenkovic I, Riggi N, Rheinbay E, Monje M, Rozenblatt-Rosen O, Cahill DP, Patel AP, Hunter T, Verma IM, Ligon KL, Louis DN, Regev A, Bernstein BE, Tirosh I, Suvà ML. An Integrative Model of Cellular States, Plasticity, and Genetics for Glioblastoma. *Cell*. 2019 Aug 8;178(4):835-849.e21. Doi: 10.1016/j.cell.2019.06.024. Epub 2019 Jul 18. PMID: 31327527; PMCID: PMC6703186.

9 A Dose Escalation Phase I Study Of Human- Recombinant Bone Morphogenetic Protein 4 Administrated Via CED In GBM Patients. Available from: <https://ClinicalTrials.gov/show/NCT02869243>.

RE: MSB-2020-10105R-Q, Modeling glioblastoma heterogeneity as a network of cell states

Thank you again for submitting your work to Molecular Systems Biology. I apologize for the delay in getting back to you with a decision on your manuscript. Unfortunately, despite repeated reminders, reviewer #2 has not yet sent us a report. In the interest of time, we have decided to proceed with making a decision based on the two available reports from reviewers #1 and #3. As you will see below, the reviewers acknowledge that the presented findings seem potentially interesting. They raise however a series of concerns, mostly related to the conclusiveness of the main findings, which preclude the publication of the study in its current form. We have decided to offer you the chance to perform a major revision and address the issues raised.

Without repeated all the points listed below, some of the more fundamental issues raised are the following:

- The presented conclusions need to be validated using an independent patient sample and the broader relevance of the approach and conclusions needs to be better substantiated.
- The relevance of the experimental setup and how well it reflects real tumors needs to be better supported.
- Reviewer #1 mentions that substantial additional analyses are required to better support the relevance of the proposed approach for candidate drug evaluation. The focus on temozolomide and BMP4 over alternative treatments/agents seems unclear and needs to be better explained and supported.
- Reviewer #3 points out that the proposed transitions between states need to be experimentally supported and alternative explanations for the changes in cell state abundances need to be excluded.
- The number of cell states needs to be better supported (reviewer #3) and a clear signature should be defined for each one of them (reviewer #1).
- Controls and statistical support should be provided on several instances.

All issues raised by the referees, including those summarized above, would need to be satisfactorily addressed. As you might already know, our editorial policy allows in principle a single round of major revision so it is essential to provide responses to the reviewers' comments that are as complete as possible. I understand that some of the issues raised by the reviewers require extensive additional analyses, please do let me know in case you need additional time beyond the 90 days offered.

Reviewer #1:

- Describe your understanding of the story

Larsson et al. use one patient-derived glioblastoma cell line profiling its single cell transcriptome longitudinally over 3 weeks (testing three time points: days 7, 14, 21). They link the changes in gene expression phenotypes to distinct glioblastoma cells and their progeny via a (lentiviral) barcoding strategy. Based on their findings, the authors fit a mathematical model that predicts six quantifiable cell states and their transition dynamics as well as overall hierarchy. The authors conclude that their methods can uncover time-dependent (gene expression) changes in heterogeneous tumour cells and can aid the systematic evaluation of anti-glioblastoma therapeutic interventions.

General remarks

- Are you convinced of the key conclusions?

If revised and strengthened (as suggested below), I am convinced that cell states and hierarchies can be evaluated in a variety of patient-derived glioblastoma cell models.

I am not yet convinced that the described approach can make a real difference for the quantitative evaluation of candidate drugs. The clinical complexity of glioblastoma (tumour microenvironment, nutrition and oxygen flux, brain location, blood brain/tumour barrier, tumour-removing surgery and adjuvant chemoradiotherapy, immune cell infiltration, use of steroids etc.) is not taken into account and I do not see the major advance over existing in vitro validation systems.

- Place the work in its context.

Mathematical modelling of stem cell hierarchies has been previously described using data based on in vivo glioblastoma modelling (Lan, Nature, 2017). Yet, I see potential conceptual novelty in combining the experimental in vitro approach with a mathematical simulation paradigm. Many groups in the field are using patient-derived glioblastoma cell lines combined with transcriptomics. The present study sheds new light onto the cellular plasticity in these widely-used model systems. Brain cancer and computational cancer biology researchers should be interested in this study. From my point of view (I am an experimental biologist,) the mathematical (STAG) model appears sufficiently novel and intriguing.

Major points

The rationale for the experimental setup is unclear. The authors transcriptionally trace the fate of ~2500 barcoded (individual) cells (in a 96-well microtiter plate). I understand that, at each time point, the cells are trypsinized and re-plated at a certain density? Which part of the glioblastoma patient

pathway could be optimally mimicked with this paradigm (relevance)? Why was the option of growing the cells clonally (mimicking self-renewal capacity) dismissed? Can a compelling rationale for the in vitro timeline, time points, cell densities, and 2 dimensionality of the assay be provided?

It is not entirely clear to me why the authors modelled cell transitions based on a system of first order chemical reactions. If transcriptional states equal the chemical species, what parameters would account for the species concentration? The reaction proceeds at a rate (k) that depends linearly on only one reactant concentration - how is this taken into account? Does this mean that non-linear state transitions (e.g. Lowengrub, 2010), which are likely relevant in the complex clinical situation of glioblastoma, are excluded from the simulation? This would pose a significant caveat for further development and use of the model. I also wonder why the typically rapid kinetics of a chemical reaction (seconds?) are viewed as an adequate kinetic for a gene expression-based readout measured over 3 weeks?

Lack of defined transcriptional states and data cross-validation. I suggest defining the transcriptional phenotypic signature for each of the six cellular states. Defining each of the transcriptional states (e.g., in Figure 3e) with only a few markers is inadequate and calls the general approach (single cell mRNAseq) into question. Once properly defined (e.g., as signatures), the mRNA expression profiles driving each state could be linked with publicly available bulk (TCGA) as well as single cell glioblastoma datasets such as Neftel et al., Cell, 2019. Showing that the six states and their dynamic relationship -obtained from a single 2-dimensional cell culture model in vitro- can be traced in an independent set of patient tumour cells would strongly increase my confidence in the potential conceptual advance that Larsson et al. are offering. Could the transcriptional profiles behind the different 'plasticity' states be linked to different survival probabilities (Kaplan-Meier estimator)? This could demonstrate clinical relevance.

Lack of a second (validation) model and non-cancerous control. Would the observed and simulated state-relationships be robust across different glioblastoma cell models. Would similar states and transitions be present in a non-cancerous neural progenitor cell line? How selective are the reported states for glioblastoma?

The rationale for comparing temozolomide and BMP4 is unclear. Why were these factors chosen over clinically-relevant radiation and environmental factors such as hypoxia? The sole use of 1 microgram/mL temozolomide and 1 ng/mL BMP4 is inadequate and not comparable to typical concentrations used in the field (typically, 50-fold higher concentrations). The focus on 'keeping' the cell proliferating despite treatment poses a risk of pre-biasing the results, hence leading to a circular argument about the ineffectiveness of anti-proliferative treatments. The conclusion that 1 ng/mL BMP4 increases the self-renewal capacity of glioblastoma should be experimentally tested by clonal analysis. How will the proposed model address the dose dependency of potential treatments and agents?

Minor points

The numbers of cells and seeding densities (cells per square centimetres) should be described for each time point.

A statistical methods section should be provided for the experimental part of the study.

t-SNE plots for each time point should be shown in the main figures and violin plots should be shown for each selected marker.

The 'Seurat' cell cycle normalization method should clearly be stated in the main text and caveats arising from the excluding cell cycle phase dynamics should be discussed in more detail.

A comparison of the growth and 'stemness' marker profiles between barcoded and parental cell lines should be provided. (Negative) mycoplasma testing results should be reported.

-Easily addressable points

Throughout the manuscript, several terms/phrases are not well defined and lack context. This is one reason why the rigour of reporting is currently only mediocre. Examples include: 'state dynamics', 'time-dependent transcriptional heterogeneity', 'new combinatorial approaches', 'accurate lineage tracing', 'plasticity of a GBM culture', 'rewiring of GBM plasticity', 'minimal stabilizing interventions', 'predict enhancements that further reduce the cell population', 'stability analysis'. The authors should make an effort to make the research accessible to potential readers that are neither (directly) familiar with the glioblastoma cell biology nor the cancer computational (math modelling) biology fields.

To me, it was not always clear as to whether the authors refer to experimental or math modelling data. For example, is Figure 4 based on modelling algorithms (simulation) or are these the experimental observations?

I would suggest restructuring the manuscript using 3 parts: 1 - experimental observations/single cell transcriptional profiling, 2 - math model and simulations, 3 - experimental and computational cross validation.

-Presentation and style

Figure design is suboptimal. Figure 1 is entirely descriptive and of supplementary info quality. Panels and text in Figures 3 and 4 appear small (very small labels) and space could be used much more efficiently by using the adequate width.

The narrative often contains a mix of results and discussion (i.e., pages 12-14). I suggest a clear separation between results and the discussion to present the data in a more reader-friendly format.

Was 'Biorender' used for scheme design (Fig. 1). If yes, this probably needs to be correctly acknowledged?

-Trivial mistakes

Throughout the manuscript, there are several (letter) printing artefacts (at least in my version). Example: PAGE 20; 'configuration o f g enes, a nd w here t he f future s tate o f each g ene' etc.

Reviewer #3:

The manuscript from Larsson and colleagues uses bar coding in conjunction with single cell RNAseq to examine heterogeneity within a population of patient derived GBM cells. Specifically, they construct a model by profiling barcoded GBM cells and their progeny over the course of 3 weeks and by applying first-order chemical kinetics to develop hypotheses regarding cell state transitions. Overall the approach is clear and the implementation should be fairly easy using the optimization packages they mentioned. The model is reasonable and straightforward.

However, a central concern regarding this work is that, while state changes are argued to occur between transcriptionally demarcated cell states, this is never directly demonstrated. The authors specifically predict that specific cell states transition into others with a specific hierarchy and a terminal state. They should provide direct experimental evidence for this. Specifically, they should isolate cell populations based on marker genes that characterize these states and examine how they change over time. Otherwise, the clear possibility remains is that changes in population size is due largely to differences in proliferation and death rates - not cell state transitions.

A second fundamental question that should be addressed is whether these populations actually achieve an equilibrium state - and whether this equilibrium is eventually re-established following treatment with BMP4 or TMZ. Notably, if these cells have not reached an equilibrium state, it would be interesting if their algorithm could predict future cell state compositions upon further cell culturing.

Finally, a tricky part of this work is actually determining the number of states within the cell population. Most existing methods rely on prior knowledge on how many cell states/groups there are or makes the simplifying assumption that there are only a defined number of possible states, so it's really bold they are trying to do unsupervised clustering. It would be meaningful to show whether the results of this work dramatically change if a distinct number of cell states (5 or 7) were chosen instead of 6.

Reviewer 1 - comments

1. *The rationale for the experimental setup is unclear. The authors transcriptionally trace the fate of ~2500 barcoded (individual) cells (in a 96-well microtiter plate). I understand that, at each time point, the cells are trypsinized and re-plated at a certain density? Which part of the glioblastoma patient pathway could be optimally mimicked with this paradigm (relevance)? Why was the option of growing the cells clonally (mimicking self-renewal capacity) dismissed? Can a compelling rationale for the in vitro timeline, time points, cell densities, and 2 dimensionality of the assay be provided?*

Response: Thank you for the interesting comment. We have now clarified further our rationale for the experimental setup used and specifically address why we choose a pooled barcoding approach in a 2D in vitro setting (Discussion, Appendix). A key rationale for the re-plating strategy is to obtain a random and sufficiently large sample of cells, which underlies the estimation of state transitions. Clonal expansions have been pursued by our team but do not provide a random sample, mainly because the cloning procedure itself acts as a major selection step, as a minority of cells (14-49%) will give rise to expandable cultures (Segerman et al, 2016). We have clarified these points in the manuscript (Results, p 5; Discussion, p 14) and provide an in-depth discussion of the considerations behind our system in the Appendix (p2). As a long-term development, we are currently exploring ways in which states could be tracked without detachment (using fluorescent tags) but this is reserved for future work. Concerning the choice of time-scale, we think that it is reasonable and appropriate to consider a time-scale of days, as this is a common choice when exploring differentiation in both malignant and non-malignant cells (Bhat et al 2013, Balasubramaniyan et al 2015). Since a strong point of our model is that it can simultaneously detect changes in growth and transitions, it also makes sense to sample the system at a time scale that is on the same order as that of the cell division rate. Like we explain below, we have carried out GSEA comparisons that relate the states found in our cells to normal human brain development (Eze et al 2021) and to other single-cell data (Neffel et al 2019 and others) and patient survival (TCGA), showing a correlation between cell hierarchy and developmental time, and top states and survival. These results lend credence to our approach.

Revisions:

- The theoretical rationale for random sampling emphasized more, and possible extensions to extended experimental setups (Results, p5; Discussion p14).
- Added a description of considerations behind the experimental setup (Appendix, p2)
- Extended analysis, relating our hierarchies to developmental data sets, other single cell data sets, and patient survival (Figure 3A-B, Figure 5A-B, Figure EV2).

2. *It is not entirely clear to me why the authors modelled cell transitions based on a system of first order chemical reactions. If transcriptional states equal the chemical species, what parameters would account for the species concentration? The reaction proceeds at a rate (k) that depends linearly on only one reactant concentration - how is this taken into account? Does this mean that non-linear state transitions (e.g. Lowengrub, 2010), which are likely relevant in the complex clinical situation of glioblastoma, are excluded from the simulation? This would pose a significant caveat for further development and use of the model. I also wonder why the typically rapid kinetics of a chemical reaction (seconds?) are viewed as an adequate kinetic for a gene expression-based readout measured over 3 weeks?*

Response: Thank you for the comment. We have now clarified these points and added discussion as follows. As a first clarification, we realize that our wording in the original submission might have caused some confusion, as we did not intend to equate in a literal sense the transcriptional states with chemical species. Our point was that the state dynamics described by our model can be seen in that perspective, as the mathematical description (the equations) is very similar. We have therefore

replaced the sentence with the more neutral "these rules describe a stochastic model which can be simulated, e.g. using Gillespie's method". Second, mathematical models of biological systems need to strike a balance between simplicity and complexity, which makes the best sense given the available data. The goal of our model is to estimate how cell states (defined by hundreds of genes) gradually change over time (days). The model that we propose is indeed linear in its current form, which suffices to extract interesting information on the hierarchical organization, switching between states, and therapy response. While these dynamics are well and productively described by a linear model at the time-scale studied, we acknowledge that there are aspects of the system where a non-linear model would likely be needed (Discussion, p15).

Revisions

- Use more nuanced language when relating the mathematics of our model to that of a description of first-order chemical equations (Results, p 6)
 - Further discuss that the model is linear (Discussion, p 15)
3. *Lack of defined transcriptional states and data cross-validation. I suggest defining the transcriptional phenotypic signature for each of the six cellular states. Defining each of the transcriptional states (e.g., in Figure 3e) with only a few markers is inadequate and calls the general approach (single cell mRNAseq) into question. Once properly defined (e.g., as signatures), the mRNA expression profiles driving each state could be linked with publicly available bulk (TCGA) as well as single cell glioblastoma datasets such as Neftel et al., Cell, 2019. Showing that the six states and their dynamic relationship -obtained from a single 2-dimensional cell culture model in vitro- can be traced in an independent set of patient tumour cells would strongly increase my confidence in the potential conceptual advance that Larsson et al. are offering. Could the transcriptional profiles behind the different 'plasticity' states be linked to different survival probabilities (Kaplan-Meier estimator)? This could demonstrate clinical relevance.*

Response: Thank you for the valuable comment. We have now added all the signature definitions to the Appendix and Supplementary Dataset EV1 and carried out analyses as follows. The signatures, which are based on more than 100 markers each, are compared to those of Neftel et al in Figure 3A (U3065MG) and Figure 5A (U3017MG and U3071MG). In addition, we have carried out enrichment tests against a bespoke panel of >300 Neurooncology relevant signatures (Supplementary Dataset EV2-4). For instance, the mesenchymal-like state that we find in U3065MG and U3071MG cells is more similar to the MES1 and MES2 states found by Neftel et al. We have also compared our signatures to an atlas of the early embryonic human brain, showing an association between upstream states found by our model, and markers expressed at earlier embryonic time points (Figure 5B). Like you suggest, we have estimated the activity of each of our states in the TCGA cohort. Interestingly, we find that the top state found in U3065MG and U3071MG cells are linked to shorter survival times (Cox regression, $p < 0.001$, Figure 3A and Figure EV2).

Revisions

- Signature definitions provided as Supplementary dataset EV1
 - Comparison to Neftel et al and other signatures in Figure 3A and 5A, Supplementary dataset EV2-4.
 - Comparison of our upstream and downstream states to early and late embryonic signatures in Figure 5B.
 - Comparison to survival times using Cox regression, Figure 3B and Figure EV2.
4. *Lack of a second (validation) model and non-cancerous control. Would the observed and simulated state-relationships be robust across different glioblastoma cell models. Would*

similar states and transitions be present in a non-cancerous neural progenitor cell line? How selective are the reported states for glioblastoma?

Response: Thank you for the comment. We have added two additional patients with the method (Figure 4-5), performed a systematic comparison to embryonic RNA data (Figure 5B), and performed an independent set of FACS sorting and subcloning experiments (Figure 2G-H, Figure EV1). First, in addition to the U3065MG cells (mesenchymal), we have now added analysis of U3017MG cells (classical) and U3071MG cells (mesenchymal). In all three cell lines, we find an essentially hierarchical organization, in line with the results of the first submission. In the two mesenchymal lines, we detect a mesenchymal top state, not detected in U3017MG cells. This illustrates how our method can be used to detect possible differences in GBM organization. Second, to relate to embryonic data, we performed a GSEA where our detected state signatures were compared to a new atlas of the embryonic human brain by Eze et al, which contains 58145 embryonic brain cells are annotated by Carnegie stage (embryonic time) and divided into 61 clusters. We found that our upstream states overlap more with early time points in the Eze atlas (Figure 5B). Third, to document that transitions occur, we have added FACS-based analysis and scRNAseq analysis of a clone grown from a single U3065MG cell, where we could detect all states. The results are now presented in Figure 2G-H and Figure EV1. Broadly, we agree that further studying both patient differences and relating further to developmental hierarchies is an interesting future direction, which we bring up in the Discussion.

Revisions

- Extended the experimental and computational work to analyze two more patients, confirming a similar hierarchy of cells (Figure 4-5)
- Systematic analysis of an embryonic atlas, relating upstream states to early embryonic time-points (Figure 5B).
- Extended the Discussion (p 15) to cover the possibility of extended studies.

5. *The rationale for comparing temozolomide and BMP4 is unclear. Why were these factors chosen over clinically-relevant radiation and environmental factors such as hypoxia? The sole use of 1 microgram/mL temozolomide and 1 ng/mL BMP4 is inadequate and not comparable to typical concentrations used in the field (typically, 50-fold higher concentrations). The focus on 'keeping' the cell proliferating despite treatment poses a risk of pre-biasing the results, hence leading to a circular argument about the ineffectiveness of anti-proliferative treatments. The conclusion that 1 ng/mL BMP4 increases the self-renewal capacity of glioblastoma should be experimentally tested by clonal analysis. How will the proposed model address the dose dependency of potential treatments and agents?*

Response: Thank you for the comment. We have added clarifications and discussion points as follows. First, we do think that temozolomide is well-motivated since it is a drug that is included in the standard treatment of GBM. In the original manuscript, the dose used was motivated as clinically relevant in the Methods section, as it has been found at this concentration in cerebrospinal fluid of glioblastoma patients. To clarify our motivation, this has now been moved to Results. BMP4, in turn, was chosen as it is one of the very few physiological factors known to inhibit glioblastoma cell proliferation. Consequently, there is a strong interest in this molecule in GBM stem cell research. Second, we fully agree with the reviewer that radiation and hypoxia are potentially as interesting, but we reserve the analysis of those interventions for future work. Hypoxia and radiation are now mentioned in the Discussion as future venues.

Concerning the dose of BMP4, we reasoned that it is better to err on the side of caution when using a highly potent growth factor. In the subventricular zone of mice, a dose of 0.1 ng/ml (i.e. 10% of the dose that we use) produces a strong effect on cell fate choice (Lim et al 2000). A common reason to use very high (pharmacological rather than physiological) doses of growth factors is to saturate the

receptor, typically in signal transduction studies. However, in human embryonic stem cells, a BMP4 dose of 0.5 ng/ml has been shown to produce almost the same level of signaling response as 50 ng/ml (Heemskerk et al 2019). We are aware that it is common in the glioblastoma field to use doses between 10 - 100 ng/ml. In our own work, we have found that 10 ng/ml has a significant impact on cell growth in 68% of HGCC cultures (Dalmo et al 2020) but that even the most refractory cell line display marked transcriptional changes at a concentration of 2 ng/ml (lower concentrations were not tested). In preliminary experiments in preparation for this manuscript, doses between 0.5 – 5 ng/ml rendered a partial growth inhibition of U3065MG when seeded at the cell density used in the experiment, while 10 ng/ml caused a complete stop of proliferation. Although it would have been interesting to explore the network behind such drastic changes, our own work as well as other studies have found that growth inhibition of glioblastoma cells induced by BMP4 alone is not permanent (Caren et al 2015). Thus, we deemed it more urgent to investigate network changes behind a partial growth response and thereby potentially elucidate a path towards finding combinatorial interventions, with the aim of inflicting a long-term response.

As shown in Figure 6, BMP4 has relatively complex effects on the state distribution. It tends to decrease the rapidly growing progenitor-like state 2, while also increasing state 1 and transitions towards state 5 (invasive-like state). From our work, we cannot conclude that these changes translate into altered in vitro self-renewal capacity. For instance, the U3065MG state 2 mostly resembles the top state A from the U3017MG network. In addition, when reducing cluster number to $k = 5$ in U3065MG, the previous state 1 and 2 merges into a single top state. Thus, it is tempting to speculate that cells that classify into both these states have self-renewal abilities. It would be of high interest, particularly within the GBM field, to explore the self-renewal capacity as well as other functional properties of individual states. However, this is beyond the scope of our systems biology approach and is reserved for future work.

Last, we agree that it would be highly interesting to probe further the dose-dependent changes of the network. It is, for instance, easy to foresee a generalization of our algorithm in which elements of $\delta-A$ are dose-dependent, which can be done within our convex optimization framework. We reserve this extension for future work.

Revisions

- Additional motivation for the dose of BMP4 is added on (Results, p11).
- The manuscript uses nuanced language when discussing the observed decrease of the progenitor-like state 2 and the observed increase of mesenchymal-like state 1 and invasive-like state 5 in BMP4-treated U3065MG cells, (Results, p11).
- We have added a discussion of dose-dependency as a possible extension of our model (Discussion, p15).

Reviewer 1 - minor comments

1. *The numbers of cells and seeding densities (cells per square centimetres) should be described for each time point.*

Response: This information has now been added to the Appendix.

2. *A statistical methods section should be provided for the experimental part of the study.*

Response: The statistical information that pertains specifically to the experiments is presented under each item in the experimental section. The preliminary experiments to determine the details of the experimental design and the statistics performed have now been added to the Appendix. The main statistical section comes in the computational part.

3. *t-SNE plots for each time point should be shown in the main figures and violin plots should be shown for each selected marker.*

Response: Thank you for suggesting this. We have added dimensionality reduction plots (U-MAP), that help gain some intuition for the data (Figure 2A, 2C, 4A-B). Dimensionality reduction per se is not a part of the STAG algorithm but we agree visualization can be helpful. For the original manuscript, we agree that violin plots would have been helpful to visualize the individual markers discussed. For this revision, our focus is on gene signatures rather than individual markers (Figures 4,5) making violin plots of individual markers unessential.

4. *The 'Seurat' cell cycle normalization method should clearly be stated in the main text and caveats arising from the excluding cell cycle phase dynamics should be discussed in more detail.*

Response: Thank you for the comment. This was described in the Methods part of the previous submission. We have now highlighted these aspects further in the main text (Discussion p 14) and Methods (p 18). Our setup is based on standard packages, particularly Seurat (Butler et al 2018).

5. *A comparison of the growth and 'stemness' marker profiles between barcoded and parental cell lines should be provided. (Negative) mycoplasma testing results should be reported.*

Response: First, we have compared the growth rate before and after barcoding, and find no signs that gene expression is substantially altered by barcoding (Appendix p5, Figure S1). Second, for a comparison of stemness marker profiles, we used GSEA to analyze the same set of pathways as in Figure 3, detecting no significant hits at a corresponding level of significance ($q < 0.01$). Third, we have added information on negative Mycoplasma testing, in Methods.

Reviewer 1 - easily addressable points

1. *Throughout the manuscript, several terms/phrases are not well defined and lack context. This is one reason why the rigour of reporting is currently only mediocre. Examples include: 'state dynamics', 'time-dependent transcriptional heterogeneity', 'new combinatorial approaches', 'accurate lineage tracing', 'plasticity of a GBM culture', 'rewiring of GBM plasticity', 'minimal stabilizing interventions', 'predict enhancements that further reduce the cell population', 'stability analysis'.*

Response: Thank you for pointing this out. We have carefully revised the manuscript for clarified and consistent nomenclature, with general readers in mind, adding additional details where needed.

2. *The authors should make an effort to make the research accessible to potential readers that are neither (directly) familiar with the glioblastoma cell biology nor cancer computational (math modeling) biology fields.*

Response: See above comment. The Introduction, Results, and Discussion parts of the manuscript have been revised with general readers in mind, including the figures, providing context and underlining possible applications.

3. *To me, it was not always clear as to whether the authors refer to experimental or math modelling data. For example, is Figure 4 based on modelling algorithms (simulation) or are these the experimental observations?*

Response: Thank you for pointing this out. We have now added subtitles in each figure to clarify this.

4. *I would suggest restructuring the manuscript using 3 parts: 1 - experimental observations/single cell transcriptional profiling, 2 - math model and simulations, 3 - experimental and computational cross validation.*

Response: Thank you for the suggestion. Our manuscript broadly follows this structure. Because our modeling is tightly linked to each experiment, we have preferred a structure where the data collection is followed by analysis for one cell line, followed by an extended data collection and treatment studies in more cases. We think this structure works well, in the balance. We are happy to take any Editorial guidance on further improvements.

Reviewer 1 - presentation and style

1. *Figure design is suboptimal. Figure 1 is entirely descriptive and of supplementary info quality. Panels and text in Figures 3 and 4 appear small (very small labels) and space could be used much more efficiently by using the adequate width.*

Response: We agree. We compressed and improved the info-graphics, which are now in Figure 1A. We have chosen to keep an infographic as we think this is helpful to readers. All figures have been reworked to improve font sizes, spacing, and markup.

2. *The narrative often contains a mix of results and discussion (i.e., pages 12-14). I suggest a clear separation between results and the discussion to present the data in a more reader-friendly format.*

Response: Thank you for pointing this out. We have move theln some cases, we have chosen to keep some bridging paragraphs to help the narrative.

3. *Was 'Biorender' used for scheme design (Fig. 1). If yes, this probably needs to be correctly acknowledged?*

Response: Thank you for noticing this. In the new version of the manuscript, we have used Adobe Illustrator for scheme design. Thus, BioRender is no longer being used.

4. *Throughout the manuscript, there are several (letter) printing artefacts (at least in my version). Example: PAGE 20; 'configuration o f g enes, a nd w here t he f future s tate o f each g ene' etc.*

Response: Thanks for noting this problem, which is due to a LaTeX conversion issue. This has been fixed.

Reviewer 2 comments

1. *However, a central concern regarding this work is that, while state changes are argued to occur between transcriptionally demarcated cell states, this is never directly demonstrated. The authors specifically predict that specific cell states transition into others with a specific hierarchy and a terminal state. They should provide direct experimental evidence for this. Specifically, they should isolate cell populations based on marker genes that characterize these states and examine how they change over time. Otherwise, the clear possibility remains*

is that changes in population size is due largely to differences in proliferation and death rates - not cell state transitions.

Response: Thank you for this valuable comment. To address this, we have carried out two additional, orthogonal experiments. In the first, a clonal culture of our U3065MG cell line, derived from a single cell, was subjected to single-cell RNA sequencing, which successfully detected all six states, supporting that transitions have occurred (Figure 2H). In a second experiment, we isolated U3065MG cells in State 1 by fluorescence-activated cell sorting (FACS) using three surface markers distinguishing State 1 cells (CD24-high/CD44-high/HLA-DR-low) from the other states. We found that cells successively move from state 1, displaying a broader range of marker expression levels, which indicates that cells are leaving State 1 (Figure 2G, Figure EV1). These two independent experiments provide additional support that transitions occur in GBM cells, as detected by our STAG model.

Revisions

- Added two additional experiments validating the occurrence of cell state transitions (Figure 2G and Figure 2H, EV1)
- 2. *A second fundamental question that should be addressed is whether these populations actually achieve an equilibrium state - and whether this equilibrium is eventually re-established following treatment with BMP4 or TMZ. Notably, if these cells have not reached an equilibrium state, it would be interesting if their algorithm could predict future cell state compositions upon further cell culturing.*

Response. Thank you for this interesting suggestion. We have now added an analysis of the idea of state composition equilibrium in our model, with predictions of long-term equilibria (Results page 11-12 and Methods page 20-21). Concisely, the analysis gives two main insights. First, it is possible to talk about two types of steady-state: a *numeric steady state* in which the total number of tumor cells is constant over time, and a *cell state equilibrium*, where the relative proportions among the *cell states* is fixed. In the original submission, we used eigenvalues to discuss the numerical steady state. In the revised manuscript, we now clarify that both of these are given by the eigendecomposition of the A matrix ($A=DV$, where D are eigenvalues and V are eigenvectors). According to our model, a numerical steady state occurs when all *eigenvalues* are zero on the real axis, the cell population is constant (numerical steady-state). A cell state equilibrium occurs when A has a real-valued eigenvector $v=\{v_1, v_2, \dots, v_k\}$, in which case the proportion f_i of each state i is $f_i=v_i/(v_1+v_2+\dots+v_k)$ (See Methods). Our model, fitted from experimental data, predicts that a mixture steady state exists in both treated and untreated cells. Thus, our model can be used to explore the likely effects of treatments on cell state equilibrium; these results are shown in Figure 6G, extending our earlier analysis. An extended experimental study of these predictions is reserved for future work.

Revisions

- Added analysis on state composition equilibrium (Results p 11-12, Methods p 20-21, Figure 6G)
- 3. *Finally, a tricky part of this work is actually determining the number of states within the cell population. Most existing methods rely on prior knowledge on how many cell states/groups there are or makes the simplifying assumption that there are only a defined number of possible states, so it's really bold they are trying to do unsupervised clustering. It would be meaningful to show whether the results of this work dramatically change if a distinct number of cell states (5 or 7) were chosen instead of 6.*

Response. Thank you for the comment. We agree that cluster number is an important hyperparameter of the analysis. The original submission did contain a relatively detailed discussion of

this topic in the Results and Supplementary Methods sections. The choice of $k=6$ is motivated, both in the main text and supported in Appendix Supplementary Methods. As case in point, consider the case of going from $k=5$ to $k=6$ in U3065MG cells (Figure EV3). If we set $k=5$, we find that states 1 and 2 are combined. Comparing to the STAG network in Figure 2E, this would correspond to a 'larger' top state in a network with an otherwise preserved structure. For $k=7$, the major change is that state 5 is split in two, which again wouldn't alter the network notably. Similar data is provided for the additional cell lines U3017MG and U3071MG (Figure EV4). As a second point, we would like to stress that in brain tumor research, and in cancer research overall, it has indeed been a common approach to employ unsupervised clustering to identify transcriptional subgroups. Key examples from the GBM field include the classical studies by Phillips et al, Verhaak et al, and more recent work by Neftel et al. So assigning transcriptional subgroups by clustering, followed by characterization of pathways and markers, is non-controversial.

Revisions

- We have added a discussion on the use of clustering methods and motivation of our choice (Discussion, p14).

Comments by the Editor

- *Please provide 5 keywords.*

DONE. Now added on page 1.

- *Please provide a .doc version of the manuscript text (including legends for main figures) and individual production quality figure files for the main Figures (one file per figure).*

RESPONSE. The manuscript is prepared in LaTeX, we have attached a full set of LaTeX files. We hope this will be acceptable for production. We will be happy to provide a Word conversion if necessary but think that the equations are likely best handled in the LaTeX format.

- *We have replaced Supplementary Information by the Expanded View (EV format). In this case, all additional figures can be included in a PDF called Appendix. Appendix figures and Tables should be labeled and called out as: "Appendix Figure S1, Appendix Figure S2... Appendix Table S1..." etc. Each legend should be below the corresponding Figure/Table in the Appendix. Please include a Table of Contents in the beginning of the Appendix. For detailed instructions regarding expanded view please refer to our Author Guidelines: <http://msb.embopress.org/authorguide#expandedview>.*

DONE. We have arranged the supplementary figures and text into an Appendix, as requested.

- *Please provide a "standfirst text" summarizing the study in one or two sentences (approximately 250 characters), three to four "bullet points" highlighting the main findings and a "synopsis image" (550px width and max 400px height, jpeg format) to highlight the paper on our homepage.*

DONE. These were attached as a separate Word document.

- *All Materials and Methods need to be described in the main text. We would encourage you to use 'Structured Methods', our new Materials and Methods format. According to this format, the Material and Methods section should include a Reagents and Tools Table (listing key reagents, experimental models, software and relevant equipment and including their sources and relevant identifiers) followed by a Methods and Protocols section in which we encourage*

the authors to describe their methods using a step-by-step protocol format with bullet points, to facilitate the adoption of the methodologies across labs. More information on how to adhere to this format as well as downloadable templates (.doc or .xls) for the Reagents and Tools Table can be found in our author guidelines:

<https://www.embopress.org/page/journal/17444292/authorguide#textformat>. An example of a Method paper with Structured Methods can be found here:

DONE. A complete Materials and Methods is found in the main text.

- Please include a Data availability section describing how the data and code have been made available. This section needs to be formatted according to the example below: The datasets and computer code produced in this study are available in the following databases:

*- Chip-Seq data: Gene Expression Omnibus GSE46748
(<https://www.ncbi.nlm.nih.gov/geo/query/acc.cgi?acc=GSE46748>)*

*- Modeling computer scripts: GitHub
(<https://github.com/SysBioChalmers/GECKO/releases/tag/v1.0>)*

- [data type]: [full name of the resource] [accession number/identifier] ([doi or URL or identifiers.org/DATABASE:ACCESSION])

DONE. A data availability section has been added.

- The file "Model Scripts" should be provided as "Computer Code EV1".

DONE. The code is now provided as a GitHub-repository.

- Please format the References according to the Molecular Systems Biology reference style.

DONE.

- For data quantification: please specify the name of the statistical test used to generate error bars and P values, the number (n) of independent experiments (specify technical or biological replicates) underlying each data point and the test used to calculate p-values in each figure legend. The figure legends should contain a basic description of n, P and the test applied. Graphs must include a description of the bars and the error bars (s.d., s.e.m.).

DONE. Statistical details are provided.

*- When you resubmit your manuscript, please download our CHECKLIST (<https://bit.ly/EMBOPressAuthorChecklist>) and include the completed form in your submission. *Please note* that the Author Checklist will be published alongside the paper as part of the transparent process (<https://www.embopress.org/page/journal/17444292/authorguide#transparentprocess>).*

DONE. Checklists attached

References

Segerman A, Niklasson M, Haglund C, Bergström T, Jarvius M, Xie Y, Westermark A, Sönmez D, Hermansson A, Kastemar M, Naimaie-Ali Z, Nyberg F, Berglund M, Sundström M, Hesselager G,

Uhrbom L, Gustafsson M, Larsson R, Fryknäs M, Segerman B, Westermark B. Clonal Variation in Drug and Radiation Response among Glioma-Initiating Cells Is Linked to Proneural-Mesenchymal Transition. *Cell Rep.* 2016 Dec 13;17(11):2994-3009. doi: 10.1016/j.celrep.2016.11.056. PMID: 27974212.

Bhat KPL, Balasubramaniyan V, Vaillant B, Ezhilarasan R, Hummelink K, Hollingsworth F, Wani K, Heathcock L, James JD, Goodman LD, Conroy S, Long L, Lelic N, Wang S, Gumin J, Raj D, Kodama Y, Raghunathan A, Olar A, Joshi K, Pelloski CE, Heimberger A, Kim SH, Cahill DP, Rao G, Den Dunnen WFA, Boddeke HWGM, Phillips HS, Nakano I, Lang FF, Colman H, Sulman EP, Aldape K. Mesenchymal differentiation mediated by NF- κ B promotes radiation resistance in glioblastoma. *Cancer Cell.* 2013 Sep 9;24(3):331-46. doi: 10.1016/j.ccr.2013.08.001. Epub 2013 Aug 29. PMID: 23993863; PMCID: PMC3817560.

Balasubramaniyan V, Vaillant B, Wang S, Gumin J, Butalid ME, Sai K, Mukheef F, Kim SH, Boddeke HW, Lang F, Aldape K, Sulman EP, Bhat KP, Colman H. Aberrant mesenchymal differentiation of glioma stem-like cells: implications for therapeutic targeting. *Oncotarget.* 2015 Oct 13;6(31):31007-17. doi: 10.18632/oncotarget.5219. PMID: 26307681; PMCID: PMC4741584.

Eze UC, Bhaduri A, Haeussler M, Nowakowski TJ, Kriegstein AR. Single-cell atlas of early human brain development highlights heterogeneity of human neuroepithelial cells and early radial glia. *Nat Neurosci.* 2021 Apr;24(4):584-594. doi: 10.1038/s41593-020-00794-1. Epub 2021 Mar 15. PMID: 33723434; PMCID: PMC8012207.

Neftel C, Laffy J, Filbin MG, Hara T, Shore ME, Rahme GJ, Richman AR, Silverbush D, Shaw ML, Hebert CM, Dewitt J, Gritsch S, Perez EM, Gonzalez Castro LN, Lan X, Druck N, Rodman C, Dionne D, Kaplan A, Bertalan MS, Small J, Pelton K, Becker S, Bonal D, Nguyen QD, Servis RL, Fung JM, Mylvaganam R, Mayr L, Gojo J, Haberler C, Geyeregger R, Czech T, Slavc I, Nahed BV, Curry WT, Carter BS, Wakimoto H, Brastianos PK, Batchelor TT, Stemmer-Rachamimov A, Martinez-Lage M, Frosch MP, Stamenkovic I, Riggi N, Rheinbay E, Monje M, Rozenblatt-Rosen O, Cahill DP, Patel AP, Hunter T, Verma IM, Ligon KL, Louis DN, Regev A, Bernstein BE, Tirosh I, Suvà ML. An Integrative Model of Cellular States, Plasticity, and Genetics for Glioblastoma. *Cell.* 2019 Aug 8;178(4):835-849.e21. doi: 10.1016/j.cell.2019.06.024. Epub 2019 Jul 18. PMID: 31327527; PMCID: PMC6703186.

Lim DA, Tramontin AD, Trevejo JM, Herrera DG, García-Verdugo JM, Alvarez-Buylla A. Noggin antagonizes BMP signaling to create a niche for adult neurogenesis. *Neuron.* 2000 Dec;28(3):713-26. doi: 10.1016/s0896-6273(00)00148-3. PMID: 11163261.

Heemskerk I, Burt K, Miller M, Chhabra S, Guerra MC, Liu L, Warmflash A. Rapid changes in morphogen concentration control self-organized patterning in human embryonic stem cells. *Elife.* 2019 Mar 4;8:e40526. doi: 10.7554/eLife.40526. PMID: 30829572; PMCID: PMC6398983.

Dalmo E, Johansson P, Niklasson M, Gustavsson I, Nelander S, Westermark B. Growth-Inhibitory Activity of Bone Morphogenetic Protein 4 in Human Glioblastoma Cell Lines Is Heterogeneous and Dependent on Reduced SOX2 Expression. *Mol Cancer Res.* 2020 Jul;18(7):981-991. doi: 10.1158/1541-7786.MCR-19-0638. Epub 2020 Mar 31. PMID: 32234828.

Carén H, Stricker SH, Bulstrode H, Gargica S, Johnstone E, Bartlett TE, Feber A, Wilson G, Teschendorff AE, Bertone P, Beck S, Pollard SM. Glioblastoma Stem Cells Respond to Differentiation Cues but Fail to Undergo Commitment and Terminal Cell-Cycle Arrest. *Stem Cell Reports.* 2015 Nov 10;5(5):829-842. doi: 10.1016/j.stemcr.2015.09.014. PMID: 26607953; PMCID: PMC4649264. Phillips HS, Kharbanda S, Chen R, Forrest WF, Soriano RH, Wu TD, Misra A, Nigro JM, Colman H, Soroceanu L, Williams PM, Modrusan Z, Feuerstein BG, Aldape K. Molecular subclasses of high-

grade glioma predict prognosis, delineate a pattern of disease progression, and resemble stages in neurogenesis. *Cancer Cell*. 2006 Mar;9(3):157-73. doi: 10.1016/j.ccr.2006.02.019. PMID: 16530701.

Verhaak RG, Hoadley KA, Purdom E, Wang V, Qi Y, Wilkerson MD, Miller CR, Ding L, Golub T, Mesirov JP, Alexe G, Lawrence M, O'Kelly M, Tamayo P, Weir BA, Gabriel S, Winckler W, Gupta S, Jakkula L, Feiler HS, Hodgson JG, James CD, Sarkaria JN, Brennan C, Kahn A, Spellman PT, Wilson RK, Speed TP, Gray JW, Meyerson M, Getz G, Perou CM, Hayes DN; Cancer Genome Atlas Research Network. Integrated genomic analysis identifies clinically relevant subtypes of glioblastoma characterized by abnormalities in PDGFRA, IDH1, EGFR, and NF1. *Cancer Cell*. 2010 Jan 19;17(1):98-110. doi: 10.1016/j.ccr.2009.12.020. PMID: 20129251; PMCID: PMC2818769.

RE: MSB-2020-10105RR, Modeling glioblastoma heterogeneity as a dynamic network of cell states

Thank you for sending us your revised manuscript. We have now heard back from the two reviewers who were asked to evaluate your study. Overall, both reviewers think that the study has improved as a result of the performed revisions. As you will see below, reviewer #1 still raises some remaining concerns, which we would ask you to address in a minor revision.

We would also ask you to address some editorial issues listed below.

Reviewer #1:

The authors have addressed most of my technical concerns and I feel that the quality of the manuscript and the displayed data (including additional computational cross-validation and additional experiments), figures and narrative have improved. The STAG model and algorithms now appear sound within the in vitro experimental context and the model system that the authors describe. In terms of potential clinical relevance, there may be clear limitations associated with the chosen approach:

-The lack of 3D environmental context and functional/in vivo validation omitting clinical considerations (such as margin zone dynamics post surgery, radiation therapy and response, immune phenotype etc.) should be clarified in a dedicated limitations paragraph (e.g., at the end of the discussion).

-Couturier et al., (Nat Comm, 2020) should be cited/discussed as the findings of this study (using single cell mRNAseq from patient specimens in combination with in vivo validation of a the 'top-of-hierarchy' progenitor cell state). Accordingly, should the statement: 'first method to quantitatively model cell state changes in solid tumor cells under normal growth and therapeutic intervention' be removed (or at least rephrased?). If possible, I suggest to include the 'Couturier_progenitor_state' in the STAG computational cross-validation - are there similarities (i.e., within the 'top-of-hierarchy' transcriptional states)?

-Can the author's clarify whether the use of STAG (as a download from github) will be limited to adherent glioma models and is a lentiviral barcoding and (3-week) in vitro cell growth strategy recommended to validate/use STAG in the wider community?

Reviewer #3:

The revised manuscript is substantially improved, and support publication of this work.

Responses to reviewer 1

The authors have addressed most of my technical concerns and I feel that the quality of the manuscript and the displayed data (including additional computational cross-validation and additional experiments), figures and narrative have improved. The STAG model and algorithms now appear sound within the in vitro experimental context and the model system that the authors describe. In terms of potential clinical relevance, there may be clear limitations associated with the chosen approach:

1. *-The lack of 3D environmental context and functional/in vivo validation omitting clinical considerations (such as margin zone dynamics post surgery, radiation therapy and response, immune phenotype etc.) should be clarified in a dedicated limitations paragraph (e.g., at the end of the discussion).*

Response: Thank you for the suggestion. We have extended the discussion of limitations to cover all these particular aspects, and formatted it into a separate paragraph in the discussion (page 14).

2. *Couturier et al., (Nat Comm, 2020) should be cited/discussed as the findings of this study (using single cell mRNAseq from patient specimens in combination with in vivo validation of a the 'top-of-hierarchy' progenitor cell state). Accordingly, should the statement: 'first method to quantitatively model cell state changes in solid tumor cells under normal growth and therapeutic intervention' be removed (or at least rephrased?). If possible, I suggest including the 'Couturier_progenitor_state' in the STAG computational cross-validation - are there similarities (i.e., within the 'top-of-hierarchy' transcriptional states)?*

Response: Thank you for the comment. We have added a citation to this particular paper and included the glial progenitor-like signature in the cross-validation (Results, p8 and Figure 3A and 5A). We have paraphrased/nuanced the sentence to "a new method to model mathematically the cell state changes in solid tumor cells under normal growth and therapeutic intervention".

RE: MSB-2020-10105RRR, Modeling glioblastoma heterogeneity as a dynamic network of cell states

Thank you again for sending us your revised manuscript. We are now satisfied with the modifications made and I am pleased to inform you that your paper has been accepted for publication.

Corresponding Author Name: Sven Nelander

Journal Submitted to: Molecular Systems Biology (MSB)

Manuscript Number: MSB-2020-10105RR